# On Sparse Modern Hopfield Model

**Jerry Yao-Chieh Hu**[†]    **Donglin Yang**[†]    **Dennis Wu**[†]

**Chenwei Xu**[†]    **Bo-Yu Chen**[‡]    **Han Liu**[†♮]

[†]Department of Computer Science, Northwestern University, Evanston, IL 60208 USA
[‡]Department of Physics, National Taiwan University, Taipei 10617, Taiwan
[♮]Department of Statistics and Data Science, Northwestern University, Evanston, IL 60208 USA

{jhu, dlyang, hibb, cxu}@u.northwestern.edu
b12202023@ntu.edu.tw, hanliu@northwestern.edu

## Abstract

We introduce the sparse modern Hopfield model as a sparse extension of the modern Hopfield model. Like its dense counterpart, the sparse modern Hopfield model equips a memory-retrieval dynamics whose one-step approximation corresponds to the sparse attention mechanism. Theoretically, our key contribution is a principled derivation of a closed-form sparse Hopfield energy using the convex conjugate of the sparse entropic regularizer. Building upon this, we derive the sparse memory retrieval dynamics from the sparse energy function and show its one-step approximation is equivalent to the sparse-structured attention. Importantly, we provide a sparsity-dependent memory retrieval error bound which is provably tighter than its dense analog. The conditions for the benefits of sparsity to arise are therefore identified and discussed. In addition, we show that the sparse modern Hopfield model maintains the robust theoretical properties of its dense counterpart, including rapid fixed point convergence and exponential memory capacity. Empirically, we use both synthetic and real-world datasets to demonstrate that the sparse Hopfield model outperforms its dense counterpart in many situations.

## 1   Introduction

We address the computational challenges of modern Hopfield models by introducing a sparse Hopfield model. Our sparse continuous Hopfield model equips a memory-retrieval dynamics that aligns with the sparse-structured attention mechanism. By establishing a connection to sparse attention, the proposed model not only offers a theoretically-grounded energy-based model for associative memory but also enables robust representation learning and seamless integration with deep learning architectures. This approach serves as an initial attempt of pushing the correspondence[1] between Hopfield models and attention mechanism [Ramsauer et al., 2021] toward sparse region, both theoretically and empirically, resulting in data-dependent sparsity for meaningful and robust pattern representations, and a focus on the most relevant information for each specific instance.

Hopfield models are classic associative memory models for both biological and artificial neural networks [Hopfield, 1982, 1984]. These models are designed to store and retrieve memory patterns. They achieve these by embedding the memories in the energy landscape of a physical system (e.g., the Ising model in [Hopfield, 1982, Peretto and Niez, 1986]; see Figure 3 for a visualization), where each memory corresponds to a local minimum. When a query is presented, the model initiates energy-minimizing retrieval dynamics at the query, which then navigate the energy landscape to find the nearest local minimum, effectively retrieving the memory most similar to the query.

---

Code is available at GitHub; future updates are on arXiv. [Version: Novermber, 28, 2023]

[1]While this equivalence only holds when the retrieval dynamics is applied exactly once, as originally shown in [Ramsauer et al., 2021] and later emphasized in [Krotov and Hopfield, 2021], it allows us to view modern Hopfield models as generalized attentions with additional functionalities and hence opens new avenues for Hopfield-based architecture designs. See Appendix C for more discussions.

In the same vein, Ramsauer et al. [2021] propose the modern Hopfield model and integrate it into deep learning architectures via a strong connection with transformer attention, offering enhanced performance, theoretically guaranteed exponential memory capacity, and the ability to handle continuous patterns. In addition, the modern Hopfield models have found success in various applications, such as immunology [Widrich et al., 2020] and large language model [Fürst et al., 2022]. Apart from the elegant connection to attention, theoretical advantages and empirically successes, the modern Hopfield models have been shown to be computationally heavy and vulnerable against noisy queries [Millidge et al., 2022]. In particular, the dense output alignments of the retrieval dynamics in modern Hopfield models [Ramsauer et al., 2021] can be computationally inefficient, making models less interpretable and noise-sensitive by assigning probability mass to many implausible outputs (patterns/keys).

To combat above, incorporating sparsity is an essential and common strategy. While there is a vast body of work on sparsifying attention mechanisms [Tay et al., 2022, Beltagy et al., 2020, Qiu et al., 2019, Child et al., 2019, Peters et al., 2019, Martins and Astudillo, 2016], similar developments for the Hopfield models remain less explored. To bridge this gap, we present a sparse Hopfield model that corresponds to the sparsemax attention mechanism [Martins and Astudillo, 2016]. In this paper, we study the sparsification of the modern Hopfield model. The challenges are three-fold:

(C1) **Non-Trivial Sparsification — Sparse Hopfield ↔ Sparse Attention:** To enable the use of sparse Hopfield models as computational devices (DNN learning models) akin to [Ramsauer et al., 2021], it is essential to achieve *non-trivial* sparsifications that exhibit equivalence to specific sparse attention models. In other words, any meaningful sparsification should extend the established equivalence [Ramsauer et al., 2021] between modern Hopfield models and attention to encompass the sparse domain. While generalizing such equivalence is potentially impactful as it may lay the groundwork for future Hopfield-based methodologies, architecture designs and bio-computing systems (as in [Kozachkov et al., 2023]), the *heuristic* design of the modern Hopfield model poses great difficulty to developing desired sparse models.

(C2) **Introducing Sparsity into Hopfield Models:** Unlike attention mechanisms where sparsification is typically achieved either on the attention matrix (e.g., structured-sparsity [Tay et al., 2020, Child et al., 2019]) or on the element-wise normalization map (e.g., sparsity-inducing maps [Correia et al., 2019, Peters et al., 2019, Martins and Astudillo, 2016]), the sparsification of Hopfield models is applied to *both* the energy function and the memory-retrieval dynamics, where the latter monotonically decreases the Hopfield energy over time. Since attention mechanisms (transformers) are typically not equipped with such a dynamical description, introducing sparsity into Hopfield models while retaining the connection to attention is a less straightforward process.

(C3) **Properties of the Sparse Hopfield Model:** Further, it is unclear how the introduced sparsity may affect different aspects of the model, such as memory capacity, fixed point convergence, retrieval accuracy, and so on. Ideally, we are looking for sparsities that offer provable computational benefits, such as enhanced robustness and increased memory capacity, among others.

Challenges (C1) and (C2) are inherent in Hopfield model, and certain requirements on the design of energy function and retrieval dynamics are inevitable to obtain non-trivial sparse models. Hence, we suppose the sparsified models should satisfy some conditions and verify them accordingly. Concretely, a formulation for deriving desired sparse Hopfield energy via convex conjugation of entropic regularizers is proposed. Furthermore, by applying Danskin's theorem and convex-concave procedure [Yuille and Rangarajan, 2003, 2001] on the sparse Hopfield energy function, we obtain sparse retrieval dynamics linked to sparse attention. For (C3), the convergence of energy stationary points and retrieval dynamics fixed points are connected via Zangwill's method [Zangwill, 1969]. The sparse retrieval error bound is derived and used to determined the well-separation condition for successful memory storage and retrieval. Lastly, the fundamental limit of memory capacity is derived using the expected separation of random points on spheres [Cai and Jiang, 2012, Brauchart et al., 2018, Ramsauer et al., 2021].

In summary, this work handles sparsification of modern Hopfield models while linking them to sparse attention by addressing the following question:

> Is it possible to develop a theoretically-grounded (non-trivial) sparse Hopfield model capable of storing information or learned prototypes throughout various layers of DNN models?

**Contributions.** We propose the Sparse Modern Hopfield Model. Our contributions are as follows:

- We propose a novel sparse Hopfield model whose retrieval dynamics corresponds to sparsemax attention mechanism. It leads to sparse patterns by design, inheriting both noise robustness and potential computational efficiency[2] from [Martins and Astudillo, 2016], compared to its dense counterparts. This work extends the theoretical understanding of the correspondence between artificial and biological neural networks to sparse region. In addition, the sparse Hopfield layer, a new deep learning component, is introduced with data-dependent sparsity.

- Theoretically, we establish provably advantages from sparsity and identify the conditions under which these benefits arise. We begin by deriving the closed-form sparse Hopfield energy from the convex conjugation of sparse entropic regularizer. Next, we demonstrate the correspondence between sparse Hopfield retrieval dynamics and sparsemax attention. In addition, we prove the fast convergence of the fixed points (also known as memory patterns, attractor states in literature) for the retrieval dynamics and establish the exponential (in pattern size) memory capacity lower bound with *tighter* retrieval error bound, *compared* with modern Hopfield models.

- Empirically, we conduct synthetic and realistic experiments to verify our theoretical results and proposed methodology. Specifically, the sparse Hopfield model outperforms the dense Hopfield model and machine learning baselines in *sparse* Multiple Instance Learning (MIL), time series prediction and neural machine translation problems. This is observed with both *sparse* synthetic and real-world datasets, where the baselines tend to fall short. Moreover, even in cases without data sparsity, our proposed model delivers performance on par with its dense counterpart.

To the best of our knowledge, we are the first to propose a sparse Hopfield model whose retrieval dynamics is equivalent to sparse attention mechanism with provably computational advantages. Methodologically, the proposed model complements existing Hopfield-based DNN architectures [Hoover et al., 2023, Paischer et al., 2022, Seidl et al., 2022, Fürst et al., 2022, Ramsauer et al., 2021] by introducing a sparse Hopfield layer into deep learning models.

**Organization.** In Section 2, the sparse Hopfield model is introduced. In Section 3, the memory capacity is discussed. In Section 4, experimental studies are conducted. In Section 5, concluding discussions are provided. Additionally, related works and limitations are discussed in Appendix C.

**Notations.** We write $\langle \mathbf{a}, \mathbf{b} \rangle \coloneqq \mathbf{a}^\mathsf{T}\mathbf{b}$ as the inner product for vectors $\mathbf{a}, \mathbf{b} \in \mathbb{R}^d$. The index set $\{1, \cdots, I\}$ is denoted by $[I]$, where $I \in \mathbb{N}_+$. The spectral norm is denoted by $\|\cdot\|$, which is equivalent to the $l_2$-norm when applied to a vector. Throughout this paper, we denote the memory patterns (keys) by $\boldsymbol{\xi} \in \mathbb{R}^d$ and the state/configuration/query pattern by $\mathbf{x} \in \mathbb{R}^d$, and $\boldsymbol{\Xi} \coloneqq (\boldsymbol{\xi}_1, \cdots, \boldsymbol{\xi}_M) \in \mathbb{R}^{d \times M}$ as shorthand for stored memory (key) patterns $\{\boldsymbol{\xi}_\mu\}_{\mu \in [M]}$. Moreover, we set norm $n \coloneqq \|\mathbf{x}\|$ be the norm of the query pattern, and $m \coloneqq \mathrm{Max}_{\mu \in [M]} \|\boldsymbol{\xi}_\mu\|$ be the largest norm of memory patterns. We also provide a nomenclature table (Table 3) in the appendix.

## 2 Sparse Hopfield Model

In this section, we introduce the sparse Hopfield energy from convex conjugate of entropic regularizer, and then the sparse retrieval dynamics. In this paper we only consider the Gini entropic regularizer corresponding to the sparsemax distribution [Martins and Astudillo, 2016].

Let $\mathbf{x} \in \mathbb{R}^d$ represent the query pattern, and let $\boldsymbol{\Xi} \coloneqq (\boldsymbol{\xi}_1, \cdots, \boldsymbol{\xi}_M) \in \mathbb{R}^{d \times M}$ denote the memory patterns. The objective of the Hopfield models is to store the memory patterns $\boldsymbol{\Xi}$ and then retrieve a specific memory pattern $\boldsymbol{\xi}_\mu$ based on a given query $\mathbf{x}$. Consequently, any Hopfield model consist of two main components: an *energy function* $\mathcal{H}(\mathbf{x})$, encoding memories into its local minima, and a *retrieval dynamics* $\mathcal{T}(\mathbf{x})$, which retrieves a memory by iteratively minimizing $\mathcal{H}(\mathbf{x})$ when initialized with a query. We provide a visualization of this procedure in Figure 3. The construction of the energy function $\mathcal{H}(\mathbf{x})$ is straightforward. As emphasized in [Krotov and Hopfield, 2016], the memories can be easily encoded into $\mathcal{H}(\mathbf{x})$ through the *overlap-construction*: $\mathcal{H}(\mathbf{x}) = F(\boldsymbol{\Xi}^\mathsf{T}\mathbf{x})$, where $F : \mathbb{R}^M \to \mathbb{R}$ is a smooth function. This ensures that the memories $\{\boldsymbol{\xi}_\mu\}_{\mu \in [M]}$ are located at the stationary points of $\mathcal{H}(\mathbf{x})$, since $\boldsymbol{\nabla}_{\mathbf{x}} F(\boldsymbol{\Xi}^\mathsf{T}\mathbf{x})|_{\boldsymbol{\xi}_\mu} = 0$ for all $\mu \in [M]$. Different choices of $F$ lead to different Hopfield models, as demonstrated in [Krotov and Hopfield, 2016, Demircigil et al., 2017, Ramsauer et al., 2021, Krotov and Hopfield, 2021]. However, finding a corresponding

---

[2]Note that, the proposed model's sparsity falls under the category of *sparsity-inducing normalization maps*. Consequently, the forward pass still requires $\mathcal{O}(n^2)$ space complexity. Here, "*potential* computational efficiency" refers that the computational efficiency can be enhanced if one employs efficient implementations that leverage sparsity, such as sort operations or median-finding algorithms, to circumvent unnecessary computations, see Appendix C and [Martins and Astudillo, 2016, Section 2] for more discussions.

retrieval dynamics, $\mathcal{T}$, for a given energy $\mathcal{H}(\mathbf{x})$, is generally more challenging. This is because $\mathcal{T}$ needs to satisfy two conditions to ensure successful memory retrieval: (i) To ensure consistent retrieval, an appropriate $\mathcal{T}$ should monotonically minimize $\mathcal{H}(\mathbf{x})$ when iteratively applied. (ii) To ensure accurate retrieval, an appropriate $\mathcal{T}$ should align its fixed points (the points where iterative application terminates) with the stationary points of $\mathcal{H}(\mathbf{x})$.

To this end, we introduce the sparse Hopfield model, providing a principled construction for $\mathcal{H}$ and $\mathcal{T}$. This model not only fulfills the aforementioned desirable properties, but also enables more robust and faster memory retrieval compared to the modern Hopfield model [Ramsauer et al., 2021].

## 2.1 Sparse Hopfield Energy

Let $\mathbf{x} \in \mathbb{R}^d$ be the query pattern, and $\mathbf{\Xi} := (\boldsymbol{\xi}_1, \cdots, \boldsymbol{\xi}_M) \in \mathbb{R}^{d \times M}$ be the memory patterns. We introduce the sparse Hopfield energy as

$$\mathcal{H}(\mathbf{x}) = -\Psi^\star \left( \beta \mathbf{\Xi}^\mathsf{T} \mathbf{x} \right) + \frac{1}{2} \langle \mathbf{x}, \mathbf{x} \rangle , \qquad (2.1)$$

with $\Psi^\star(\mathbf{z}) := \frac{1}{2}\|\mathbf{z}\|^2 - \frac{1}{2}\|\mathrm{Sparsemax}(\mathbf{z}) - \mathbf{z}\|^2 + \frac{1}{2}$, where $\mathrm{Sparsemax}(\cdot)$ is defined as follows. Let $\mathbf{z}, \mathbf{p} \in \mathbb{R}^M$, and $\Delta^M := \{\mathbf{p} \in \mathbb{R}_+^M \mid \sum_\mu^M p_\mu = 1\}$ be the $(M-1)$-dimensional unit simplex.

**Definition 2.1** (Sparsemax in Variational Form [Martins and Astudillo, 2016], also see Remark F.1)**.**
$$\mathrm{Sparsemax}(\mathbf{z}) := \underset{\mathbf{p} \in \Delta^M}{\mathrm{ArgMin}} \|\mathbf{p} - \mathbf{z}\|^2 = \underset{\mathbf{p} \in \Delta^M}{\mathrm{ArgMax}} \left[ \mathbf{p}^\mathsf{T}\mathbf{z} - \Psi(\mathbf{p}) \right] , \qquad (2.2)$$
where $\Psi(\mathbf{p}) := -\frac{1}{2}\sum_\nu^M p_\nu(1 - p_\nu)$ is the negative Gini entropy or Gini entropic regularizer.

**Remark 2.1.** Recall that, the variational form (2.2) is in fact general, that applies to various entropic regularizers, as discussed in [Peters et al., 2019, Wainwright et al., 2008]. The choice of $\Psi$ determines the resulting sparse probability distribution. For instance, if we choose the Gibbs' entropic regularizer $\Psi_{\mathrm{Gibbs}} = -\sum_\nu^M p_\nu \ln p_\nu$, (2.2) reduces to the standard softmax distribution.

**Overview of Theoretical Results.** At first glance, the energy function (2.1) may seem peculiar. However, it indeed represents a non-trivial sparse Hopfield model with appealing properties, including:

(i) In response to challenge (C1) & (C2), as we shall see in Section 2.2, the energy (2.1) leads to a sparse retrieval dynamics that not only retrieves memory by monotonically decreasing (Lemma 2.1) to its stationary points (Lemma 2.2), but also associates with sparsemax attention through its single-step approximation (Remark 2.2);

(ii) In response to challenge (C3), as we shall see in Section 3, it indulges fast convergence of retrieval (Corollary 3.1.2), exponential-in-$d$ memory capacity akin to modern Hopfield models (Lemma 3.1). Notably, it accomplishes these with a tighter retrieval error bound (Theorem 2.1).

We reveal each of these properties in the following sections.

## 2.2 Sparse Retrieval Dynamics and Connection to Sparse Attention

The optimization problem $\mathrm{ArgMax}_{\mathbf{p} \in \Delta^M} \left[ \mathbf{p}^\mathsf{T}\mathbf{z} - \Psi(\mathbf{p}) \right]$ does not necessarily have a closed-form solution for arbitrary $\Psi$. However, a family of $\Psi$ has been investigated in literature [Correia et al., 2019, Martins and Astudillo, 2016] with closed-form solutions derived, including the $\mathrm{Sparsemax}(\cdot)$.

**Sparsemax in Closed-Form** (Proposition 1 of [Martins and Astudillo, 2016])**.** Let $\mathbf{z} \in \mathbb{R}^M$. Denote $[a]_+ := \mathrm{Max}\{0, a\}$, $z_{(\nu)}$ the $\nu$'th element in a sorted descending $z$-sequence $\mathbf{z}_{\mathrm{sorted}} := z_{(1)} \geq z_{(2)} \geq \ldots \geq z_{(M)}$, and $\kappa(\mathbf{z}) := \mathrm{Max}\left\{ k \in [M] \mid 1 + kz_{(k)} > \sum_{\nu \leq k} z_{(\nu)} \right\}$. The optimization problem(s) (2.2) has closed-form solution
$$\mathrm{Sparsemax}(\mathbf{z}) = [\mathbf{z} - \tau(\mathbf{z})\mathbf{1}_M]_+ , \qquad (2.3)$$
where $\tau : \mathbb{R}^M \to \mathbb{R}$ is the threshold function $\tau(\mathbf{z}) = \left[ \left( \sum_{\nu \leq \kappa(\mathbf{z})} z_{(\nu)} \right) - 1 \right] / \kappa(\mathbf{z})$, satisfying $\sum_{\mu=1}^M [z_\mu - \tau(\mathbf{z})]_+ = 1$ for all $\mathbf{z}$. Notably, $\kappa(\mathbf{z}) = |S(\mathbf{z})|$ where $S(\mathbf{z}) = \{\mu \in [M] \mid \mathrm{Sparsemax}_\mu(\mathbf{z}) > 0\}$ is the support set of $\mathrm{Sparsemax}(\mathbf{z})$.

In this case, we present the following theorem to derive the convex conjugate of $\Psi$ in closed-form:

**Theorem 2.1** (Convex Conjugate of Negative Gini Entropy). Let $F(\mathbf{p}) := \langle \mathbf{p}, \mathbf{z} \rangle - \Psi(\mathbf{p})$ with $\Psi$ being the negative Gini entropy, $\Psi(\mathbf{p}) = \frac{1}{2}\|\mathbf{p}\|^2 - \frac{1}{2}$. The convex conjugate of $\Psi(\mathbf{p})$ is

$$\Psi^\star(\mathbf{z}) := \underset{\mathbf{p} \in \Delta^M}{\mathrm{Max}}\, F(\mathbf{p}, \mathbf{z}) = \frac{1}{2}\|\mathbf{z}\|^2 - \frac{1}{2}\|\mathbf{p}^\star - \mathbf{z}\|^2 + \frac{1}{2}, \qquad (2.4)$$

where $\mathbf{p}^\star = \mathrm{Sparsemax}(\mathbf{z})$ is given by (2.3).

**Corollary 2.1.1.** By Danskin's Theorem, $\boldsymbol{\nabla}\Psi^\star(\mathbf{z}) = \mathrm{ArgMax}_{\mathbf{p} \in \Delta^M} F(\mathbf{p}, \mathbf{z}) = \mathrm{Sparsemax}(\mathbf{z})$.

*Proof.* A detailed proof is shown in Appendix E.1. □

Theorem 2.1 and Corollary 2.1.1 not only provide the intuition behind the sparse Hopfield energy (2.1) — the memory patterns are stored in local minima aligned with the overlap-function constructions (i.e. $\|\boldsymbol{\Xi}^\mathsf{T}\mathbf{x}\|^2 = \sum_{\mu=1}^M \langle \boldsymbol{\xi}_\mu, \mathbf{x} \rangle^2$) in [Ramsauer et al., 2021, Demircigil et al., 2017, Krotov and Hopfield, 2016] — but also prepare us for the following corresponding sparse retrieval dynamics.

**Lemma 2.1** (Sparse Retrieval Dynamics). Let $t$ be the iteration number. The energy (2.1) can be monotonically decreased by the following sparse retrieval dynamics over $t$:

$$\mathcal{T}(\mathbf{x}_t) := \boldsymbol{\nabla}_\mathbf{x}\Psi\left(\beta\boldsymbol{\Xi}^\mathsf{T}\mathbf{x}\right)\Big|_{\mathbf{x}_t} = \boldsymbol{\Xi}\,\mathrm{Sparsemax}\left(\beta\boldsymbol{\Xi}^\mathsf{T}\mathbf{x}_t\right) = \mathbf{x}_{t+1}. \qquad (2.5)$$

*Proof Sketch.* To show monotonic decreasing property, we first derive the sparse retrieval dynamics by utilizing Theorem 2.1, Corollary 2.1.1, along with the convex-concave procedure [Yuille and Rangarajan, 2003, 2001]. Then, we show the monotonicity of $\mathcal{H}$ by constructing a iterative upper bound of $\mathcal{H}$ which is convex in $\mathbf{x}_{t+1}$ and thus, can be lowered iteratively by the convex-concave procedure. A detailed proof is shown in the Appendix E.2. □

**Remark 2.2.** Similar to [Ramsauer et al., 2021], (2.5) is equivalent to sparsemax attention [Martins and Astudillo, 2016] when the $\mathcal{T}$ is applied only once, see Appendix D for more details. Importantly, $\beta$ acts as a scaling factor for the energy function, often referred to as the "inverse temperature". It influences the sharpness of energy landscape Equation (2.1), thereby controlling the dynamics. High $\beta$ values, corresponding to low temperatures, encourage that the basins of attraction for individual memory patterns remain distinct, leading to easier retrieval.

Notably, since $\|\boldsymbol{\Xi}^\mathsf{T}\mathbf{x}\|^2 = \sum_{\mu=1}^M \langle \boldsymbol{\xi}_\mu, \mathbf{x} \rangle^2$, (2.5) implies that the local optimum of $\mathcal{H}$ are located near the patterns $\boldsymbol{\xi}_\mu$. Different from previous studies on binary Hopfield models [Demircigil et al., 2017, Krotov and Hopfield, 2016], for continuous patterns, we adopt the relaxed definition from [Ramsauer et al., 2021][3] to rigorously analyze the memory retrieval, and the subsequent lemma.

**Definition 2.2** (Stored and Retrieved). Assuming that every pattern $\boldsymbol{\xi}_\mu$ surrounded by a sphere $S_\mu$ with finite radius $R := \frac{1}{2}\mathrm{Min}_{\mu \neq \nu; \mu,\nu \in [M]}\|\boldsymbol{\xi}_\mu - \boldsymbol{\xi}_\nu\|$, we say $\boldsymbol{\xi}_\mu$ is *stored* if there exists a generalized fixed point of $\mathcal{T}$, $\mathbf{x}_\mu^\star \in S_\mu$, to which all limit points $\mathbf{x} \in S_\mu$ converge to, and $S_\mu \cap S_\nu = \emptyset$ for $\mu \neq \nu$. We say $\boldsymbol{\xi}_\mu$ is *$\epsilon$-retrieved* by $\mathcal{T}$ with $\mathbf{x}$ for an error[a] $\epsilon$, if $\|\mathcal{T}(\mathbf{x}) - \boldsymbol{\xi}_\mu\| \leq \epsilon$.

---
[a] The retrieval error has a naive bound $\epsilon := \mathrm{Max}\left\{\|\mathbf{x} - \boldsymbol{\xi}_\mu\|, \|\boldsymbol{\xi}_\mu - \mathbf{x}_\mu^\star\|\right\}$ by interpolating from $\mathbf{x}$ to $\boldsymbol{\xi}_\mu$.

Definition 2.2 sets the threshold for a memory pattern $\boldsymbol{\xi}_\mu$ to be considered *stored* at a fixed point of $\mathcal{T}$, $\mathbf{x}_\mu^\star$. However, this definition does not imply that the fixed points of $\mathcal{T}$ are also stationary points of the energy function $\mathcal{H}$. In fact, monotonicity of (2.5) does not assure the existence of stationary points of energy $\mathcal{H}$ [Sriperumbudur and Lanckriet, 2009]. To establish a well-defined Hopfield model, we need two types of convergence guarantees. The first is the convergence between $\mathbf{x}_\mu^\star$ and $\boldsymbol{\xi}_\mu$, which ensures that the retrieved memory is close to the stored memory. The second is the convergence of $\mathcal{H}$ to its stationary points through the dynamics of $\mathcal{T}$, which ensures that the system reaches a state of minimal energy. The following lemma provides the convergence results for both.

**Lemma 2.2** (Convergence of Retrieval Dynamics $\mathcal{T}$). Suppose $\mathcal{H}$ is given by (2.1) and $\mathcal{T}(\mathbf{x})$ is given by (2.5). For any sequence $\{\mathbf{x}_t\}_{t=0}^\infty$ defined by $\mathbf{x}_{t'+1} = \mathcal{T}(\mathbf{x}_{t'})$, all limit points of this sequence are stationary points if they are obtained by iteratively applying $\mathcal{T}$ to $\mathcal{H}$.

---
[3] Recall that a fixed point of $\mathcal{T}$ with respect to $\mathcal{H}$ is a point where $\mathbf{x} = \mathcal{T}(\mathbf{x})$, and a generalized fixed point is a point where $\mathbf{x} \in \mathcal{T}(\mathbf{x})$. For more details, refer to [Sriperumbudur and Lanckriet, 2009].

*Proof Sketch.* We verify and utilize Zangwill's global convergence theory [Zangwill, 1969] for iterative algorithms $\mathcal{T}$, to first show that all the limit points of $\{\mathbf{x}_t\}_{t=0}^{\infty}$ are generalized fixed points and $\lim_{t \to \infty} \mathcal{H}(\mathbf{x}_t) = \mathcal{H}(\mathbf{x}^\star)$, where $\mathbf{x}^\star$ are some generalized fixed points of $\mathcal{T}$. Subsequently, by [Sriperumbudur and Lanckriet, 2009, Lemma 5], we show that $\{\mathbf{x}^\star\}$ are also stationary points of $\mathrm{Min}_{\mathbf{x}}[\mathcal{H}]$, and hence $\mathcal{H}$ converges to local optimum. A detailed proof is shown in Appendix E.4. $\square$

Intuitively, Lemma 2.2 indicates that the energy function converges to local optimum, i.e. $\lim_{t \to \infty} \mathcal{H}(\mathbf{x}_t) \to \mathcal{H}(\mathbf{x}^\star)$, where $\mathbf{x}^\star$ are stationary points of $\mathcal{H}$. Consequently, it offers formal justifications for the retrieval dynamics (2.5) to retrieve stored memory patterns $\{\boldsymbol{\xi}_\mu\}_{\mu \in [M]}$: for any query (initial point) $\mathbf{x}$, $\mathcal{T}$ monotonically and iteratively approaches stationary points of $\mathcal{H}$, where the memory patterns $\{\boldsymbol{\xi}_\mu\}_{\mu \in [M]}$ are stored. As for the retrieval error, we provide the following theorem stating that $\mathcal{T}$ achieves a lower retrieval error compared to its dense counterpart.

**Theorem 2.2** (Retrieval Error). Let $\mathcal{T}_{\mathrm{Dense}}$ be the retrieval dynamics of the dense modern Hopfield model [Ramsauer et al., 2021]. It holds $\|\mathcal{T}(\mathbf{x}) - \boldsymbol{\xi}_\mu\| \le \|\mathcal{T}_{\mathrm{Dense}}(\mathbf{x}) - \boldsymbol{\xi}_\mu\|$ for all $\mathbf{x} \in S_\mu$. Moreover,

$$\|\mathcal{T}(\mathbf{x}) - \boldsymbol{\xi}_\mu\| \le m + d^{1/2} m \beta \left[ \kappa \left( \mathrm{Max}_{\nu \in [M]} \langle \boldsymbol{\xi}_\nu, \mathbf{x} \rangle - \left[\boldsymbol{\Xi}^\mathsf{T}\mathbf{x}\right]_{(\kappa)} \right) + \frac{1}{\beta} \right], \qquad (2.6)$$

where $\left[\boldsymbol{\Xi}^\mathsf{T}\mathbf{x}\right]_{(\kappa)}$ is the $\kappa$th-largest element of $\boldsymbol{\Xi}^\mathsf{T}\mathbf{x} \in \mathbb{R}^M$ following the sparsemax definition (2.3).

*Proof.* A detailed proof is shown in Appendix E.3. $\square$

Interestingly, (2.6) is a sparsity dependent bound[4]. By denoting $n := \|\mathbf{x}\|$, the second term on the RHS of (2.6) is dominated by the sparsity dimension $\kappa$ as it can be expressed as $\kappa \left(1 - \left[\boldsymbol{\Xi}^\mathsf{T}\mathbf{x}\right]_{(\kappa)}/(nm)\right) \propto \alpha\kappa$ with a constant $0 \le \alpha \le 2$. When $\boldsymbol{\Xi}^\mathsf{T}\mathbf{x}$ is sparse (i.e. $\kappa$ is small), the bound is tighter, vice versa.

**Remark 2.3** (Faster Convergence). Computationally, Theorem 2.2 implies that $\mathcal{T}$ requires fewer iterations to reach fixed points with the same amount of error tolerance compared to $\mathcal{T}_{\mathrm{dense}}$. Namely, $\mathcal{T}$ retrieves stored memory patterns faster and therefore more efficiently, as evidenced in Figure 2.

**Remark 2.4** (Noise-Robustness). Moreover, in cases of contaminated patterns with noise $\boldsymbol{\eta}$, i.e. $\widetilde{\mathbf{x}} = \mathbf{x} + \boldsymbol{\eta}$ (noise in query) or $\widetilde{\boldsymbol{\xi}}_\mu = \boldsymbol{\xi}_\mu + \boldsymbol{\eta}$ (noise in memory), the impact of noise $\boldsymbol{\eta}$ on the sparse retrieval error (2.6) is linear, while its effect on the dense retrieval error (2.7) is exponential. This suggests the robustness advantage of the sparse Hopfield model, as evidenced in Figure 1.

### 2.3 Sparse Hopfield Layers for Deep Learning

The sparse Hopfield model can serve as a versatile component for deep learning frameworks, given its continuity and differentiability with respect to parameters. Corresponding to three types of Hopfield Layers proposed in [Ramsauer et al., 2021], we introduce their sparse analogs: **(1)** `SparseHopfield`, **(2)** `SparseHopfieldPooling`, **(3)** `SparseHopfieldLayer`. Layer `SparseHopfield` has memory (stored or key) patterns $\boldsymbol{\Xi}$ and query (state) pattern $\mathbf{x}$ as inputs, and associates these two sets of patterns via the sparse retrieval dynamics (2.5). This layer regards the transformer attention layer as its one-step approximation, while utilizing the sparsemax [Martins and Astudillo, 2016] on attention matrix. Layer `SparseHopfieldPooling` and Layer `SparseHopfieldLayer` are two variants of `SparseHopfield`, whose input patterns are memory patterns and query patterns from previous layers or external plugin, respectively. `SparseHopfieldPooling`, whose query patterns are learnable parameters, can be interpreted as performing a pooling operation over input memory patterns. `SparseHopfieldLayer`, by contrast, has learnable memory patterns that maps query patterns to hidden states with sparsemax activation. Thus it can substitute a fully connected layer within deep learning architectures. See (D.12) and the implementation Algorithm 1 in Appendix D, and [Ramsauer et al., 2021, Section 3] for more details of these associations. In Section 4, we apply these layers and compare them with their dense counterparts in [Ramsauer et al., 2021] and other baseline machine learning methods.

## 3 Fundamental Limits of Memory Capacity of Sparse Hopfield Models

How many patterns can be stored and reliably retrievable in the proposed model? We address this by decomposing it into to two sub-questions and answering them separately:

(A) What is the condition for a pattern $\boldsymbol{\xi}_\mu$ considered well stored in $\mathcal{H}$, and correctly retrieved?

---

[4]Notably, $\|\mathcal{T}(\mathbf{x}) - \boldsymbol{\xi}_\mu\|$ is also upper-bounded by a sparsity-independent but $M, \beta$-dependent bound

$$\|\mathcal{T}(\mathbf{x}) - \boldsymbol{\xi}_\mu\| \le \|\mathcal{T}_{\mathrm{Dense}}(\mathbf{x}) - \boldsymbol{\xi}_\mu\| \le 2m(M-1)\exp\left\{-\beta\left(\langle \boldsymbol{\xi}_\mu, \mathbf{x} \rangle - \mathrm{Max}_{\nu \in [M], \nu \ne \mu} \langle \boldsymbol{\xi}_\mu, \boldsymbol{\xi}_\nu \rangle \right)\right\}. \qquad (2.7)$$

(B) What is the number, in expectation, of the the patterns satisfying such condition?

For (A), we first introduce the notion of separation of patterns following [Ramsauer et al., 2021],

**Definition 3.1** (Separation of Patterns). The separation of a memory pattern $\boldsymbol{\xi}_\mu$ from all other memory patterns $\boldsymbol{\Xi}$ is defined as its minimal inner product difference to any other patterns:

$$\Delta_\mu := \underset{\nu, \nu \neq \mu}{\mathrm{Min}} \left[ \langle \boldsymbol{\xi}_\mu, \boldsymbol{\xi}_\mu \rangle - \langle \boldsymbol{\xi}_\mu, \boldsymbol{\xi}_\nu \rangle \right] = \langle \boldsymbol{\xi}_\mu, \boldsymbol{\xi}_\mu \rangle - \underset{\nu, \nu \neq \mu}{\mathrm{Max}} \left[ \langle \boldsymbol{\xi}_\mu, \boldsymbol{\xi}_\nu \rangle \right]. \tag{3.1}$$

Similarly, the separation of $\boldsymbol{\xi}_\mu$ at a given $\mathbf{x}$ from all memory patterns $\boldsymbol{\Xi}$ is given by

$$\widetilde{\Delta}_\mu := \underset{\nu, \nu \neq \mu}{\mathrm{Min}} \left[ \langle \mathbf{x}, \boldsymbol{\xi}_\mu \rangle - \langle \mathbf{x}, \boldsymbol{\xi}_\nu \rangle \right]. \tag{3.2}$$

and then the well-separation condition for a pattern being well-stored and retrieved.

**Theorem 3.1** (Well-Separation Condition). Given the definition of stored and retrieved memories in Definition 2.2, suppose the memory patterns $\{\boldsymbol{\xi}_\mu\}_{\mu \in [M]}$ are located within the sphere $S_\mu := \{\mathbf{x} \mid \|\mathbf{x} - \boldsymbol{\xi}_\mu\| \leq R\}$, where the radius $R$ is finite and defined in Definition 2.2 for all $\mu$. Then, the retrieval dynamics $\mathcal{T}$ maps the sphere $S_\mu$ onto itself under the following conditions:
1. The initial query $\mathbf{x}$ is located within the sphere $S_\mu$, i.e., $\mathbf{x} \in S_\mu$.
2. The *well-separation* condition is satisfied, which is given by:

$$\Delta_\mu \geq mn + 2mR - \left[ \boldsymbol{\Xi}^\mathsf{T} \mathbf{x} \right]_{(\kappa)} - \frac{1}{\kappa} \left( \frac{R - m - md^{1/2}}{m\beta d^{1/2}} \right).$$

**Corollary 3.1.1.** Let $\delta := \|\mathcal{T}_{\mathrm{Dense}} - \boldsymbol{\xi}_\mu\| - \|\mathcal{T} - \boldsymbol{\xi}_\mu\|$. The well-separation condition can be expressed as $\Delta_\mu \geq \frac{1}{\beta} \ln \left( \frac{2(M-1)m}{R+\delta} \right) + 2mR$, which reduces to that of the dense Hopfield model when $\delta = 0$.

*Proof Sketch.* The proofs proceed by connecting $\Delta_\mu$ with $\|\mathcal{T}(\mathbf{x}) - \boldsymbol{\xi}_\mu\|$. To do so, we utilize Theorem 2.2 to incorporate the $\Delta_\mu$-dependent bound on the retrieval error of both sparse and dense Hopfield models [Ramsauer et al., 2021]. A detailed proof is shown in Appendix E.5. $\qquad\square$

Together with Lemma 2.2, the well-separated condition serves as the necessary condition for pattern $\boldsymbol{\xi}_\mu$ to be well-stored at the stationary points of $\mathcal{H}$, and can be retrieved with at most $\epsilon = R$ by $\mathcal{T}$, as per Definition 2.2. We make the following three observations about the blessings from sparsity.

1. In general, to appreciate the blessings of sparsity, we rearrange the well-separation condition as

$$\Delta_\mu \geq 2mR + \underbrace{\left( mn - \left[ \boldsymbol{\Xi}^\mathsf{T} \mathbf{x} \right]_{(\kappa)} \right)}_{:=\alpha nm \text{ with } 0 \leq \alpha \leq 2} - \frac{1}{\kappa} \left( \frac{R - m - md^{1/2}}{m\beta d^{1/2}} \right), \tag{3.3}$$

and observe the two competing terms, $\alpha nm$ and $(R-m-md^{1/2})/(\kappa m\beta d^{1/2})$. Sparsity proves advantageous when the latter term surpasses the former, i.e. the sparse well-separation condition is consistently lower than its dense counterpart. The condition under which sparsity benefits are more likely to emerge (i.e., when the well-separation condition is more readily satisfied) is thereby:

$$\frac{1}{2} \underset{\mu, \nu \in [M]}{\mathrm{Min}} \|\boldsymbol{\xi}_\mu - \boldsymbol{\xi}_\nu\| \geq md^{1/2} \left( 1 + \alpha \beta nm\kappa \right) + m, \quad \text{with } 0 \leq \alpha \leq 2. \tag{3.4}$$

Intuitively, the sparser $\boldsymbol{\Xi}^\mathsf{T} \mathbf{x}$ is, the easier it is for the above condition to be fulfilled.

2. **Large $M$ limit:** For large $M$, the dense well-separation condition (Corollary 3.1.1) explodes while the sparse one (Theorem 3.1) saturates to the first three $M$-independent terms. This suggests that the hardness of distinguishing patterns can be tamed by the sparsity, preventing an increase of $\Delta_\mu$ with $M$ as observed in the dense Hopfield model. We numerically confirm this in Figure 1.

3. **$\beta \to \infty$ Limit:** In the region of low temperature, where $\beta \to \infty$ and hence all patterns can be *error-free* retrieved as per (2.7), we have $\Delta_\mu \geq 2mR + \alpha nm$ with $0 \leq \alpha \leq 2$. Here, the second term on the RHS represents the sparsity level of $\boldsymbol{\Xi}^\mathsf{T} \mathbf{x}$, i.e. a smaller $\alpha$ indicates a higher degree of sparsity in $\boldsymbol{\Xi}^\mathsf{T} \mathbf{x}$. Hence, the higher the sparsity, the easier it is to separate patterns.

For (B), equipped with Theorem 3.1 and Corollary 3.1.1, we provide a lower bound for the number of patterns being well-stored and can be *at least $R$*-retrieved in the next lemma[5]:

---

[5]Following the convention in memory capacity literature [Ramsauer et al., 2021, Demircigil et al., 2017, Krotov and Hopfield, 2016], we assume that all memory patterns $\{\boldsymbol{\xi}_\mu\}$ are sampled from a $d$-sphere of radius $m$.

**Lemma 3.1** (Memory Capacity Lower Bound). Let $1 - p$ be the probability of successfully storing and retrieving a pattern. The number of patterns randomly sampled from a sphere of radius $m$ that the sparse Hopfield model can store and retrieve is lower-bounded by

$$M \geq \sqrt{p}C^{\frac{d-1}{4}}, \tag{3.5}$$

where $C$ is the solution to $C = {^b/_{W_0(\exp\{a + \ln b\})}}$ with $W_0(\cdot)$ being the principal branch of Lambert $W$ function, $a := (4/_{d-1})\left(\ln\left[2m(\sqrt{p}-1)/(R+\delta)\right] + 1\right)$ and $b := {^{4m^2\beta}/_{5(d-1)}}$. For sufficiently large $\beta$, the sparse Hopfield model exhibits a larger lower bound on the exponential memory capacity compared to its dense counterpart [Ramsauer et al., 2021]: $M \geq M_{\text{Dense}}$.

*Proof Sketch.* Our proof is built on [Ramsauer et al., 2021]. The high-level idea is to utilize the separation of random patterns sampled from spheres [Cai and Jiang, 2012, Brauchart et al., 2018] and the asymptotic expansion of the Lambert $W$ function [Corless et al., 1996]. Firstly, we link the well-separation condition to cosine similarity distance, creating an inequality for the probability of a pattern being well-stored and retrieved. Next, we identify and prove conditions for the exponential memory capacity $M = \sqrt{p}C^{(d-1)/4}$ to hold. Finally, we analyze the scaling behaviors of $C$ using its asymptotic expansion and show that $M \geq M_{\text{Dense}}$. A detailed proof is shown in Appendix E.6. □

Intuitively, the benefits of sparsity arises from the increased energy landscape separation provided by the sparse Hopfield energy function, which enables the separation of closely correlated patterns, resulting in a tighter well-separation condition for distinguishing such patterns and hence a larger lower bound on the memory capacity. Moreover, the sparse Hopfield model also enjoys the properties of fast convergence and exponentially suppressed retrieval error provided by the following corollary.

**Corollary 3.1.2** (Fast Convergence and Exponentially Suppressed Retrieval Error). For any query $\mathbf{x}$, $\mathcal{T}$ approximately retrieves a memory pattern $\boldsymbol{\xi}_\mu$ with retrieval error $\epsilon$ exponentially suppressed by $\Delta_\mu$: $\|\mathcal{T}(\mathbf{x}) - \boldsymbol{\xi}_\mu\| \leq 2m(M-1)\exp\left\{-\beta\left(\Delta_\mu - 2m\,\text{Max}\left[\|\mathbf{x} - \boldsymbol{\xi}_\mu\|, \|\mathbf{x} - \mathbf{x}_\mu^\star\|\right]\right)\right\}$.

*Proof.* This results from Theorem 2.2, Lemma 2.2, and [Ramsauer et al., 2021, Theorem 4]. □

Corollary 3.1.2 suggests that, with a sufficient $\Delta_\mu$, $\mathcal{T}$ can approximately retrieve patterns after a single *activation*, allowing the integration of sparse Hopfield models into deep learning architectures similarly to [Hoover et al., 2023, Seidl et al., 2022, Fürst et al., 2022, Ramsauer et al., 2021].

## 4 Proof of Concept Experimental Studies

We demonstrate the validity of our theoretical results and method by testing them on various experimental settings with both synthetic and real-world datasets.

### 4.1 Experimental Validation of Theoretical Results

We conduct experiments to verify our theoretical findings, and report the results in Figure 1. For the memory capacity (the top row of Figure 1), we test the proposed sparse model on retrieving half-masked patterns comparing with the Dense (Softmax) and 10th order polynomial Hopfield models [Millidge et al., 2022, Krotov and Hopfield, 2016] on MNIST (high sparsity), Cifar10 (low sparsity) and ImageNet (low sparsity) datasets. For all Hopfield models, we set $\beta = 1$.[6] A query is regarded as correctly retrieved if its cosine similarity error is below a set threshold. In addition, for the robustness against noisy queries (the bottom row of Figure 1), we inject Gaussian noises with varying variances ($\sigma$) into the images. Plotted are the means and standard deviations of 10 runs. The results show that the proposed sparse Hopfield model excels when memory patterns exhibit a high degree of sparsity and the signal-to-noise ratio in patterns is low, aligning with our theoretical results.

### 4.2 Multiple Instance Learning Tasks

Ramsauer et al. [2021] point out that the memory-enhanced Hopfield layers present a promising approach for Multiple Instance Learning (MIL) tasks. Multiple Instance Learning (MIL) [Ilse et al., 2018, Carbonneau et al., 2018] is a variation of supervised learning where the training set consists of labeled bags, each containing multiple instances. The goal of MIL is to predict the bag labels based on the instances they contain, which makes it particularly useful in scenarios where labeling individual instances is difficult or impractical, but bag-level labels are available. Examples of such scenarios include medical imaging (where a bag could be an image, instances could be patches of the

---

[6]However, as pointed out in [Millidge et al., 2022], this is in fact *not* fair to compare modern Hopfield with $\beta = 1$ with higher order polynomial Hopfield models.

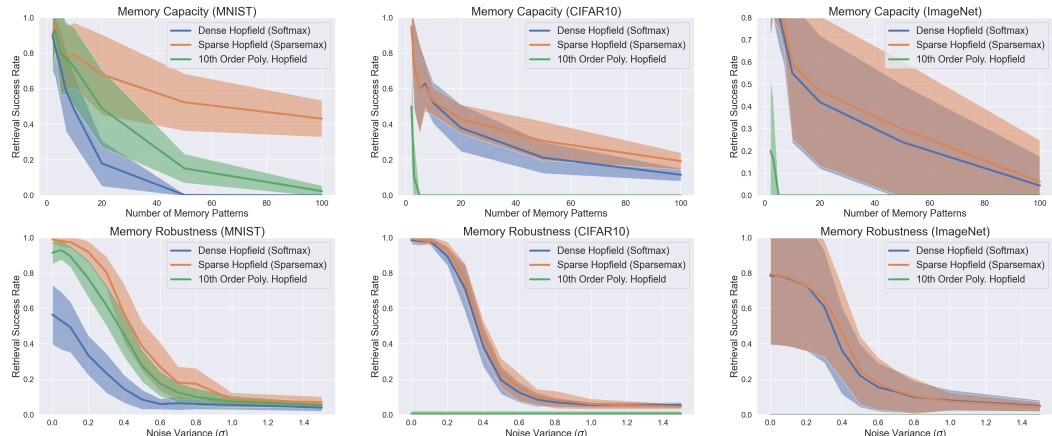

Figure 1: **Top:** Memory Capacity measured by successful half-masked retrieval rates. **Bottom:** Memory Robustness measured by retrieving patterns with varying levels of Gaussian noise. For all Hopfield models, we set $\beta = .01/0.1/0.1$ (for MNIST/CIFAR10/ImageNet) for better visualization. A query pattern is deemed correctly retrieved if its squared Euclidean distance is below a set threshold. For MNIST/CIFAR10/ImageNet datasets, we set the error thresholds to be 10/20/20 to cope with different sparse levels in data. Plotted are the means and standard deviations of 10 runs. The results suggest that the sparse Hopfield model excels when memory patterns exhibit a high degree of sparsity and the signal-to-noise ratio in patterns is low.

image, and the label could indicate the presence or absence of disease) and document classification (where a bag could be a document, instances could be the words or sentences in the document, and the label could indicate the topic or sentiment of the document). In this subsection, we implement our sparse Hopfield layers and applied them to MIL tasks on one synthetic and four real-world settings.

### 4.2.1 Synthetic Experiments

We use a synthetic MIL dataset, the bit pattern dataset, to demonstrate the effectiveness of the sparse Hopfield model. Each bag in this synthetic dataset contains a set of binary bit strings. The positive bag includes at least one of the positive bit patterns. We compare the performance of the `SparseHopfield` and `SparseHopfieldPooling` to their dense counterparts and vanilla attention [Vaswani et al., 2017]. We report the mean test accuracy of 10 runs. To demonstrate the effectiveness of sparse Hopfield model, we vary two hyperparameters of the bit pattern dataset corresponding to two perspectives: bag sparsity (sparsity in data) and bag size (number of memory patterns, $M$). For **bag sparsity**, we fix the bag size as 200, and inject from 2 to 80 positive patterns in a positive bag, results in 1 to 40 percent of positive patterns in each positive bag. For **bag size**, we fix the number of positive pattern in a bag to be 1, and vary bag size from 20 to 300. We report results of `SparseHopfieldPooling` in Table 1, and implementation details in Appendix H.1.1. A more complete version of Table 1, including the results of `Hopfield` and attention, is in Appendix G. The sparse Hopfield model demonstrates a better performance across all sparsity and all bag sizes.

Table 1: **Top (Bag Size):** Accuracy comparison on bit pattern dataset for sparse and dense Hopfield model. We report the average accuracy over 10 runs. The results suggest that the sparse Hopfield model demonstrates a better performance when facing a bag size increase. **Bottom (Bag Sparsity):** Performance comparison on bit pattern dataset for sparse and dense Hopfield model with varying bag sparsity. We report the average accuracy over 10 runs. The results suggest that the sparse Hopfield model demonstrates a better performance across all sparsity.

| Bag Size | 20 | 50 | 100 | 150 | 200 | 300 |
|---|---|---|---|---|---|---|
| Dense Hopfield Pooling | $100.0 \pm 0.00$ | $100.0 \pm 0.00$ | $100.0 \pm 0.00$ | $76.44 \pm 0.23$ | $49.13 \pm 0.01$ | $52.88 \pm 0.01$ |
| Sparse Hopfield Pooling | $100.0 \pm 0.00$ | $100.0 \pm 0.00$ | $100.0 \pm 0.00$ | $\mathbf{99.76 \pm 0.00}$ | $\mathbf{99.76 \pm 0.00}$ | $\mathbf{99.76 \pm 0.00}$ |

| Bag Sparsity | 1% | 5% | 10% | 20% | 40% |
|---|---|---|---|---|---|
| Dense Hopfield Pooling | $49.20 \pm 0.00$ | $85.58 \pm 0.10$ | $100.0 \pm 0.00$ | $100.0 \pm 0.00$ | $99.68 \pm 0.00$ |
| Sparse Hopfield Pooling | $\mathbf{73.40 \pm 0.06}$ | $\mathbf{99.68 \pm 0.00}$ | $100.0 \pm 0.00$ | $100.0 \pm 0.00$ | $\mathbf{100.0 \pm 0.00}$ |

**Convergence Analysis.** In Figure 2, we numerically examine the convergence of the sparse and dense Hopfield models, plotting their loss and accuracy for the **bag size** tasks in above on the bit pattern

dataset. We include multiple bag sizes to assess the effect of increasing memory patterns (i.e. $M$) on the loss curve. The plotted are the loss and accuracy curves of `SparseHopfieldPooling`. We refer results of `Hopfield` and more details to Appendix G.3. The results (Figure 2) show that, sparse Hopfield model surpasses its dense counterpart in all bag sizes. Moreover, for the same bag size, the sparse Hopfield model always reaches the minimum validation loss faster than dense Hopfield model, validating our Theorem 2.2.

**Sparsity Generalization.** We also evaluate the models' generalization performance with shifting information sparsity, by training dense and sparse Hopfield models with a specific bag sparsity and testing them on the other. We report the results in Table 5 and refer more details to Appendix G.3.

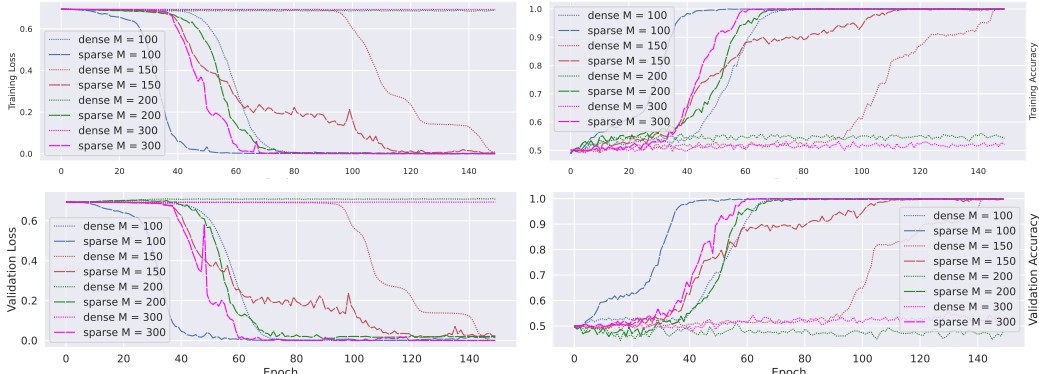

Figure 2: **Top:** The training loss and accuracy curve of dense and sparse Hopfield models with different bag sizes. **Bottom:** The validation loss and accuracy curve of dense and sparse Hopfield models with different bag sizes. The plotted are the mean of 10 runs. The results indicate that the sparse Hopfield model converges faster than the dense model and also yields superior accuracy.

### 4.2.2 Real-World MIL Tasks

Next, we demonstrate that the proposed method achieves near-optimal performance on four realistic (*non-sparse*) MIL benchmark datasets: Elephant, Fox and Tiger for image annotation [Ilse et al., 2018], UCSB breast cancer classification [Kandemir et al., 2014]. We use `Hopfield` and `SparseHopfield` to construct a similar model architecture proposed in [Ramsauer et al., 2021] and a detailed description of this experiment as well as its training and evaluating process can be found in Appendix H.1.2. As shown in Table 2, both Sparse and Dense Hopfield achieve near-best results on Tiger, Elephant and UCSB datasets, despite the low sparsity in data. The sparse Hopfield model outperforms the dense Hopfield model by a small margin on three out of four datasets.

## 5 Conclusion

We present a sparse Hopfield model with a memory-retrieval dynamics that corresponds to the sparse-structured attention mechanism. This model is capable of merging into deep learning architectures with data-dependent sparsity. Theoretically, we introduce a principled construction for modern Hopfield models, based on the convex conjugate of different entropy regularizers. It allows us to easily recover the dense modern Hopfield model [Ramsauer et al., 2021] using Gibbs entropy. Moreover, we introduce the sparse Hopfield model using the Gini entropic regularizer, and ex-

Table 2: Results for MIL benchmark datasets in terms of AUC score. The baselines are Path encoding [Küçükaşcı and Baydoğan, 2018], MInD [Cheplygina et al., 2015], MILES [Chen et al., 2006], APR [Dietterich et al., 1997], Citation-KNN [Wang and Zucker, 2000] and DD [Maron and Lozano-Pérez, 1997]. Results for baselines are taken from [Ramsauer et al., 2021]. The results suggest the proposed model achieves near-optimal performance even when the data is not sparse.

| Method | Tiger | Fox | Elephant | UCSB |
|---|---|---|---|---|
| Dense Hopfield | $0.878 \pm 0.028$ | $0.600 \pm 0.011$ | $0.907 \pm 0.022$ | $0.880 \pm 0.013$ |
| Sparse Hopfield | $0.892 \pm 0.021$ | $0.611 \pm 0.010$ | $0.912 \pm 0.016$ | $0.877 \pm 0.009$ |
| Path encoding | $0.910 \pm 0.010$ | $0.712 \pm 0.014$ | $0.944 \pm 0.007$ | $0.880 \pm 0.022$ |
| MInD | $0.853 \pm 0.011$ | $0.704 \pm 0.016$ | $0.936 \pm 0.009$ | $0.831 \pm 0.027$ |
| MILES | $0.872 \pm 0.017$ | $0.738 \pm 0.016$ | $0.927 \pm 0.007$ | $0.833 \pm 0.026$ |
| APR | $0.778 \pm 0.007$ | $0.541 \pm 0.009$ | $0.550 \pm 0.010$ | — |
| Citation-kNN | $0.855 \pm 0.009$ | $0.635 \pm 0.015$ | $0.896 \pm 0.009$ | $0.706 \pm 0.032$ |
| DD | $0.841$ | $0.631$ | $0.907$ | — |

plore its theoretical advantages, delineating conditions that favor its use. Empirically, we demonstrate our theoretical results and methodology to be effective on various synthetic and realistic settings. This work extends the correspondence between artificial and biological neural networks to sparse domain, potentially paving the way for future Hopfield-based methodologies and bio-inspired computing systems.

**Post-Acceptance Note [November 28, 2023].** After the completion of this work, the authors learn of two upcoming works — [Anonymous, 2023] at ICLR'24 and [Martins et al., 2023] in the Associative Memory & Hopfield Networks Workshop at NeurIPS'23 — both addressing similar topics. Both of these works explore theoretical generalizations of this work. In addition, [Anonymous, 2023] further presents a (sparse) Hopfield-based deep learning model for multivariate time series prediction. We thank the authors of [Martins et al., 2023] for enlightening discussions and for sharing their preliminary manuscript.

## Acknowledgments

JH would like to thank Tim Tsz-Kit Lau, Robin Luo and Andrew Chen for enlightening discussions on related topics, and Jiayi Wang for invaluable support in facilitating experimental deployments. The authors would like to thank the anonymous reviewers and program chairs for constructive comments.

JH is partially supported by the Walter P. Murphy Fellowship. HL is partially supported by NIH R01LM1372201, NSF CAREER1841569, DOE DE-AC02-07CH11359, DOE LAB 20-2261 and a NSF TRIPODS1740735. This research was supported in part through the computational resources and staff contributions provided for the Quest high performance computing facility at Northwestern University which is jointly supported by the Office of the Provost, the Office for Research, and Northwestern University Information Technology. The content is solely the responsibility of the authors and does not necessarily represent the official views of the funding agencies.

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
