# Appendix

# A  Nomenclature Table

We summarize our notations in the following table for easy reference.

Table 3: Mathematical Notations and Symbols

| Symbol | Description |
|---|---|
| $\langle \mathbf{a}, \mathbf{b} \rangle$ | Inner product for vectors $\mathbf{a}, \mathbf{b} \in \mathbb{R}^d$ |
| $[I]$ | Index set $\{1, \cdots, I\}$, where $I \in \mathbb{N}^+$ |
| $\lVert \cdot \rVert_2$ | Spectral norm, equivalent to the $l_2$-norm when applied to a vector |
| $d$ | Dimension of patterns |
| $M$ | Number of stored memory patterns |
| $\beta$ | A scaling factor of the energy function that controls the learning dynamics |
| $\mathbf{x}$ | State/configuration/query pattern in $\mathbb{R}^d$ |
| $\boldsymbol{\xi}$ | Memory patterns (keys) in $\mathbb{R}^d$ |
| $\boldsymbol{\Xi}$ | Shorthand for $M$ stored memory (key) patterns $\{\boldsymbol{\xi}_\mu\}_{\mu \in [M]}$ in $\mathbb{R}^{d \times M}$ |
| $\boldsymbol{\Xi}^\mathsf{T} \mathbf{x}$ | $M$-dimensional overlap vector $(\langle \boldsymbol{\xi}_1, \mathbf{x} \rangle, \cdots, \langle \boldsymbol{\xi}_\mu, \mathbf{x} \rangle, \cdots, \langle \boldsymbol{\xi}_M, \mathbf{x} \rangle)$ in $\mathbb{R}^M$ |
| $\left[ \boldsymbol{\Xi}^\mathsf{T} \mathbf{x} \right]_\kappa$ | The $\kappa$-th element of $\boldsymbol{\Xi}^\mathsf{T} \mathbf{x}$ |
| $n$ | Norm of $\mathbf{x}$, denoted as $n := \lVert \mathbf{x} \rVert$ |
| $m$ | Largest norm of memory patterns, denoted as $m := \mathrm{Max}_{\mu \in [M]} \lVert \boldsymbol{\xi}_\mu \rVert$ |
| $\kappa$ | The number of non-zero element of $\mathrm{Sparsemax}$, defined in (2.3) |
| $R$ | The minimal Euclidean distance across all possible pairs of memory patterns, $R := \frac{1}{2} \mathrm{Min}_{\mu, \nu \in [M]} \lVert \boldsymbol{\xi}_\mu - \boldsymbol{\xi}_\nu \rVert$ |
| $S_\mu$ | The sphere centered at the memory pattern $\boldsymbol{\xi}_\mu$ with finite radius $R$ |
| $\mathbf{x}_\mu^\star$ | The fixed point of $\mathcal{T}$ covered by $S_\mu$, i.e. $\mathbf{x}_\mu^\star \in S_\mu$ |
| $\Delta_\mu$ | The separation of a memory pattern $\boldsymbol{\xi}_\mu$ from all other memory patterns $\boldsymbol{\Xi}$, defined in (3.1) |
| $\widetilde{\Delta}_\mu$ | The separation of $\boldsymbol{\xi}_\mu$ at a given $\mathbf{x}$ from all memory patterns $\boldsymbol{\Xi}$, defined in (3.2) |

# B  Broader Impacts and Future Directions: Brain Science and Foundation Models

The primary theme of our research is to perceive any data representation (set of patterns) as analogous to the neural responses of a global brain reacting to a vast range of external stimuli (queries). This perspective presents exciting opportunities to study large generative foundational models, such as large language models, within a rigorous scientific framework inspired by contemporary brain science research.

We believe this work could be impactful in several respects, even though it is foundational research and not tied to specific applications: **(Cognition.)** This research could contribute to our understanding of a memory-enhanced model's predictive capacity when given either in-context input (like historical data) or external stimuli (such as real-time events). **(Memory.)** It may also shed light on the inherent limits of artificial neural networks' memorization capabilities and how to augment them with external memory modules for rapid responses to potential external stimuli. **(Network.)** This research could enable models to better assess the intricate network of cross-sectional brain activity among different variables and infer its dynamic structural alterations to identify possible systematic properties.

# C  Related Works and Limitations

**Sparse Hopfield Models.**   Our work is closely related to and motivated by [Földiak, 1990], which proposes a local anti-Hebbian learning rule for sparse representations in associative memory networks. This rule enhances storage capacity and retrieval capabilities but has limitations: (i) fixed sparsity based on local similarity of receptive fields, (ii) difficulty in scaling up and integration with modern DNNs [Makhzani and Frey, 2015], (iii) lack of a solid theoretical foundation for convergence and stability, and (iv) inherently unsupervised retrieval dynamics, limiting its applicability for supervised learning or other paradigms like reinforcement learning or semi-supervised learning. On the other hand, another line of related work, not specifically focusing on sparsifying Hopfield models, centers on sparse coding [Palm, 2013, Olshausen and Field, 1997], introducing sparsity to associative memory models through thresholding memory patterns. These studies offer insights into the relationship between the sparseness of the stored memory patterns and the robustness of the network but sufferers from the issues related to scalability, sparsity level bias, and noise vulnerability [Mairal et al., 2010, Rubinstein et al., 2010, Elad, 2010, Olshausen and Field, 1997]. In contrast, our approach is

theoretically grounded and has data-dependent sparsity leading to better scalability, more meaningful and robust representations of patterns and allows the model to focus on the most relevant information for each specific instance.

**Hopfield Models and Connection to Attention.** Hopfield Models [Hopfield, 1984, 1982, Krotov and Hopfield, 2016] have seen renewed interest in the machine learning community due to advances in memory storage capacity understanding [Krotov and Hopfield, 2016, Demircigil et al., 2017], architectural innovations [Hoover et al., 2023, Seidl et al., 2022, Fürst et al., 2022, Ramsauer et al., 2021], and biological plausibility [Kozachkov et al., 2023, Krotov and Hopfield, 2021]. Notably, Modern Hopfield Networks [Ramsauer et al., 2021][7], a new subclass, highlight the equivalence[8] between their memory retrieval dynamics and attention mechanisms in transformers. With this hindsight, it becomes clear that transformers and modern Hopfield models share some high-level similarities, as well as differences. Both architectures are designed for denoising input, with transformers typically pre-trained on masked-token tasks and the modern Hopfield model aimed at completing incomplete or contaminated patterns. However, the modern Hopfield models are recurrent networks with a global energy function that ensures convergence to a fixed-point attractor, while transformers are generally viewed as feed-forward networks without such dynamics. It is natural to ask whether such equivalence is fundamental. Although, apart from Hopfield-side investigations [Hoover et al., 2023, Krotov and Hopfield, 2021, Ramsauer et al., 2021], there have been studies viewing transformers as dynamical systems, including the deep equilibrium models [Bai et al., 2019], and unfolded optimization [Yang et al., 2022], none exhibit similar converge-to-memory dynamics as in Hopfield models (hence missing the connection between dynamical memory retrieval and transformers), nor do they address sparsity. Building on the established equivalence in [Ramsauer et al., 2021], our work serves as an initial attempt to push such equivalence toward sparse models, both theoretically and empirically. It lays the groundwork for future Hopfield-based methodologies, architecture designs and biological computers (as in [Kozachkov et al., 2023]).

**Sparse Attention.** Attention-based seq2seq models excel in various applications like large language models [Chowdhery et al., 2022, Brown et al., 2020], time series prediction [Zhou et al., 2022, 2021], and biomedical science [Ji et al., 2021], primarily due to their versatility in framing tasks as source-to-target sequence transformations with potentially varying lengths. However, the original transformer architecture utilizes a dense, quadratic attention score matrix, which can be computationally demanding (with $\mathcal{O}(n^2)$ complexity for input sequence length $n$), memory-intensive, and challenging to interpret for long sequences. To combat these issues, there is a large amount of literature works leverages various sparsifying methods for attention and transformers to enhance computational efficiency while preserving the models' expressiveness, see [Tay et al., 2022] for an overview. Here, we classify sparse Transformers into two distinct categories based on the different kinds of sparsities. The first category focuses on structured-sparsity [Beltagy et al., 2020, Qiu et al., 2019, Child et al., 2019], which involves creating a sparse attention score matrix in a pre-determined manner. In these approaches, each token in the sequence attends to a fixed subset of other tokens, rather than the entire sequence. The second category obtains sparsity through the sparsity-inducing normalization maps [Peters et al., 2019, Correia et al., 2019, Krotov and Hopfield, 2016] that encourage the models to focus on a subset of relevant input elements, thereby fostering sparsity, scalability and interpretability. Compared to the first category, while these approaches still have $\mathcal{O}(n^2)$ space complexity, they offer the advantage of producing sparsity patterns that are more adaptive to the data. Our work is closely related to the second and utilizes *sparsity-inducing alternatives* to the softmax function in modern Hopfield models.

## C.1 Limitations

Since our model aligns with sparsemax attention, it also grapples with $\mathcal{O}(d^2)$ complexity, a characteristic typical of the sparsity-inducing normalization map category of sparse attention. In addition, we opt not to impose any assumptions on the data (patterns) to maintain the general applicability of

---

[7]Also see the well-written blog post [Brandstetter, 2021].

[8]While this equivalence only holds when the retrieval dynamics is applied exactly once, as originally shown in [Ramsauer et al., 2021] and later emphasized in [Krotov and Hopfield, 2021], it allows us to view modern Hopfield models as generalized attentions with additional functionalities and hence opens new avenues for Hopfield-based architecture designs.

our model. This decision, however, prevents us from providing a rigorous characterization of how data-dependent sparsity explicitly impacts retrieval error, the well-separation condition, and memory capacity. Specifically, a detailed analysis of $\left[\boldsymbol{\Xi}^{\mathsf{T}}\mathbf{x}\right]_{(\kappa)}$ is a problem of order statistics [David and Nagaraja, 2004] that hinges on the distribution of patterns. Instead, we offer qualitative discussions in Section 3 to provide insights into the behavior of the sparse model under various conditions, aiding in a better understanding and application of the model.

# D  Modern Hopfield Model and Its Connection to Attention Mechanism

Ramsauer et al. [2021] generalize the exponential-interaction-based energy function proposed in [Demircigil et al., 2017] to continuous patterns and states with a strong link the attention mechanism. In this section, we provide an overview for both of them and then draw the connection of the modern Hopfield model to attention mechanism.

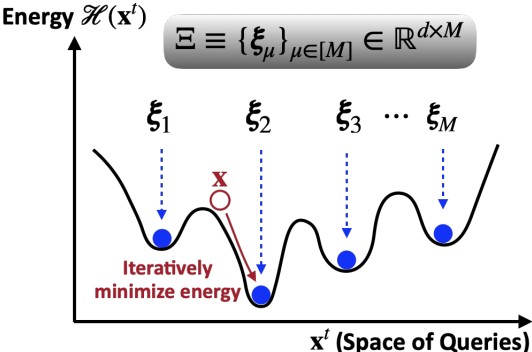

Figure 3: **Visualizing Hopfield Models.** Let $\mathbf{x} \in \mathbb{R}^d$ represent the query pattern, and let $\boldsymbol{\Xi} := (\boldsymbol{\xi}_1, \cdots, \boldsymbol{\xi}_M) \in \mathbb{R}^{d \times M}$ denote the memory patterns. The objective of the Hopfield models is to store the memory patterns $\boldsymbol{\Xi}$ and then retrieve a specific memory pattern $\boldsymbol{\xi}_\mu$ based on a given query $\mathbf{x}$. They achieve these by embedding the memories $\boldsymbol{\Xi}$ in the energy landscape $\mathcal{H}(\mathbf{x})$ of a physical system (e.g., the Ising model in [Hopfield, 1982] and its higher-order generalizations [Lee et al., 1986, Peretto and Niez, 1986, Newman, 1988]), where each memory $\boldsymbol{\xi}_\mu$ corresponds to a local minimum. When a query $\mathbf{x}$ is presented, the model initiates energy-minimizing retrieval dynamics $\mathcal{T}$ at the query $\mathbf{x}$, which then navigate the energy landscape to find the nearest local minimum, effectively retrieving the memory most similar to the query.

## D.1  Modern Hopfield Model

We first introduce the log-sum-exponential (lse) function for any given vector $\mathbf{z} = (z_1, \cdots, z_M)$ and $\beta > 0$:

$$\mathrm{lse}\,(\beta, \mathbf{z}) := \frac{1}{\beta} \log \left( \sum_{\mu=1}^{M} \exp\{\beta z_\mu\} \right), \tag{D.1}$$

which is an important representation of the softmax function which can be derived by considering the "argmax function" under entropy regularization, see [Gao and Pavel, 2017] and references therein.

**Exponential Binary Hopfield Model.**  With $\mathrm{lse}(\cdot)$, the exponential-Hopfield model for binary patterns $\boldsymbol{\xi}, \mathbf{x} \in \{\pm 1\}^d$ proposed in [Demircigil et al., 2017] can be written as, denoting $\boldsymbol{\Xi} := (\boldsymbol{\xi}^1, \cdots, \boldsymbol{\xi}^M)$,

$$\mathcal{H}(\mathbf{x}) = -\sum_{\mu=1}^{M} \exp\{\langle \boldsymbol{\xi}^\mu, \boldsymbol{\sigma} \rangle\} = -\exp\{\mathrm{lse}\left(1, \boldsymbol{\Xi}^{\mathsf{T}} \boldsymbol{\sigma}\right)\}, \tag{D.2}$$

which leads to the super-linear memory capacity of $M \propto 2^{d/2}$.

**Modern Hopfield Model.** For continuous patterns $\mathbf{x}, \{\boldsymbol{\xi}^\mu\} \in \mathbb{R}^d$, Ramsauer et al. [2021] propose the continuous[9] modern Hopfield model

$$\mathcal{H}(\mathbf{x}) \coloneqq -\operatorname{lse}\left(1, \boldsymbol{\Xi}^\mathsf{T}\mathbf{x}\right) + \frac{1}{2}\langle\mathbf{x},\mathbf{x}\rangle + \frac{1}{\beta}\log M + \frac{1}{2}m^2, \tag{D.3}$$

with retrieval dynamics

$$\mathbf{x}^{\text{new}} = \mathcal{T}_{\text{Dense}}(\mathbf{x}) = \boldsymbol{\Xi}\cdot\operatorname{Softmax}\left(\beta\boldsymbol{\Xi}^\mathsf{T}\mathbf{x}\right), \tag{D.4}$$

where $\frac{1}{2}\langle\mathbf{x},\mathbf{x}\rangle$ is a regularizer introduced for ensuring configuration vector $\mathbf{x}$ being finite, and $m \coloneqq \max_\mu \|\boldsymbol{\xi}^\mu\|$ is the largest norm of memory patterns. Moreover, they show that (i) the modern Hopfield (D.3) has an exponential memory capacity in $d$, (ii) the retrieval dynamics (D.4) can consistently retrieve patterns with high accuracy with only one step, and (iii) surprisingly, the retrieval dynamics (D.4) is connected to the attention mechanism in transformer giving rise to a new methodology — the Hopfield DNN layer.

## D.2 Memory Retrieval Dynamics $\mathcal{T}_{\text{Dense}} \leftrightarrow$ Self-Attention Mechanism

Following [Ramsauer et al., 2021, Brandstetter, 2021], we say $\mathbf{X}$ and $\boldsymbol{\Xi}$ are in the associative space (embedded space), as they are mapped from the *raw* query $\mathbf{R}$ and $\mathbf{Y}$ memory patterns, respectively, via

$$\mathbf{X}^\mathsf{T} = \mathbf{R}\mathbf{W}_Q \coloneqq \mathbf{Q}, \tag{D.5}$$

$$\boldsymbol{\Xi}^\mathsf{T} = \mathbf{Y}\mathbf{W}_K \coloneqq \mathbf{K}, \tag{D.6}$$

with some $\mathbf{W}_Q$ and $\mathbf{W}_K$. Therefore, we can express $\mathcal{T}_{\text{Dense}}$ as

$$(\mathbf{Q}^{\text{new}})^\mathsf{T} = \mathbf{K}^\mathsf{T}\operatorname{Softmax}\left(\beta\mathbf{K}\mathbf{Q}^\mathsf{T}\right). \tag{D.7}$$

Taking transpose to above, we have

$$\mathbf{Q}^{\text{new}} = \operatorname{Softmax}\left(\beta\mathbf{Q}\mathbf{K}^\mathsf{T}\right)\mathbf{K}. \tag{D.8}$$

Projecting $\mathbf{K}$ to $\mathbf{V}$ with $\mathbf{W}_V$, we have

$$\boldsymbol{Z} \coloneqq \mathbf{Q}^{\text{new}}\mathbf{W}_V = \operatorname{Softmax}\left(\beta\mathbf{Q}\mathbf{K}^\mathsf{T}\right)\mathbf{K}\mathbf{W}_V \tag{D.9}$$

$$= \operatorname{Softmax}\left(\beta\mathbf{Q}\mathbf{K}^\mathsf{T}\right)\mathbf{V}, \tag{D.10}$$

which leads to the self-attention mechanism.

Plugging back the raw patterns $\mathbf{R}$ and $\mathbf{Y}$, we arrive the foundation of the Hopfield layer,

$$\boldsymbol{Z} = \operatorname{Softmax}\left(\beta\mathbf{R}\mathbf{W}_Q\mathbf{W}_K^\mathsf{T}\mathbf{Y}^\mathsf{T}\right)\mathbf{Y}\mathbf{W}_K\mathbf{W}_V. \tag{D.11}$$

The same construction applies to the sparse retrieval dynamics (2.5),

$$\boldsymbol{Z}' = \operatorname{Sparsemax}\left(\beta\mathbf{R}\mathbf{W}_Q'\mathbf{W}_K'^\mathsf{T}\mathbf{Y}^\mathsf{T}\right)\mathbf{Y}\mathbf{W}_K'\mathbf{W}_V'. \tag{D.12}$$

resulting in a sparse Hopfield layer that can be seamlessly integrated into deep learning architectures.

## D.3 Algorithm of Multi-Step `SparseHopfield` Layer

Here, we present an algorithm for implementing the `SparseHopfield` layer with multi-step updates (i.e. multiple iterative retrievals). The algorithm, summarized in Algorithm 1 below, outlines the process for $U$ update steps. Similar to [Ramsauer et al., 2021], the `SparseHopfield` takes as input the matrices $\mathbf{R}, \mathbf{Y}$, and the weight matrices $\mathbf{W}_Q', \mathbf{W}_K', \mathbf{W}_V'$.

Here we explain the usage of the above algorithm w.r.t. different settings.

1. **Memory Retrieval.** The memory retrieval is a learning-free setting. Thus, we can exclude the use of weight matrices $\mathbf{W}_K, \mathbf{W}_Q, \mathbf{W}_V$ (by setting them to identity matrices). And let the input (corrupted image) to be our $\mathbf{R}$, stored patterns as $\mathbf{Y}$ for retrieval.

---

[9]Note that, there are also many continuous Hopfield models prior than [Ramsauer et al., 2021], including [Krotov and Hopfield, 2016, Hopfield, 1984].

**Algorithm 1** Multi-Step `SparseHopfield` Layer

---

**Require:** $U \in \mathbb{R} \geq 1, \mathbf{R}, \mathbf{Y}$.
$\quad \mathbf{Q} \leftarrow \mathbf{RW}'_Q$
$\quad$ **for** $i \rightarrow 1$ to $U$ **do**
$\quad\quad \mathbf{Q}^{\text{new}} \leftarrow \text{Sparsemax}\left(\beta \mathbf{QW}'^T_K \mathbf{Y}^T\right) \mathbf{YW}'_V \mathbf{W}'_K \qquad\qquad$ *Hopfield Update* as D.12
$\quad\quad \mathbf{Q} \leftarrow \mathbf{Q}^{\text{new}}$
$\quad$ **end for**
$\quad$ **return** $\mathbf{Q}$

---

2. `SparseHopfield`**.** The `SparseHopfield` has two inputs, $\mathbf{R}, \mathbf{Y}$. Since the `SparseHopfield` can be used to replace attention mechanism in models, we make the weight matrices $\mathbf{W}'_K, \mathbf{W}'_Q, \mathbf{W}'_V$ learnable, and $\mathbf{R}, \mathbf{Y}, \mathbf{Y}$ be the source of query, key, value, respectively. Note that the self-attention-liked mechanism can be realized by setting $\mathbf{R} = \mathbf{Y}$.

3. `SparseHopfieldPooling`**.** The `SparseHopfieldPooling` layer has one input, $\mathbf{Y}$, where $\mathbf{Q}$ is the learnable **prototype pattern** and fixed during inference, and $\mathbf{Y}$ is the stored patterns we want to perform pooling over. Note that the $\mathbf{Q}$ here is independent from the input and can be seen as part of the learnable parameter of the `SparseHopfieldPooling` layer. Here since we replace the query pattern ($\mathbf{RW}'_Q$) with a static **prototype pattern Q**, the learnable weight matrices here will only be $\mathbf{W}'_K, \mathbf{W}'_V$.

4. `SparseHopfieldLayer`**.** The `SparseHopfieldLayer` layer has one input, $\mathbf{R}$. Where $\mathbf{R}$ is the query pattern. And we have learnable weight matrices $\mathbf{W}'_K, \mathbf{W}'_V$ served as our stored patterns and pattern projections, leading our key and value independent to the input. In other words, following the notation in Algorithm 1, $\mathbf{Y}$ can be seen as an identity matrix.

# E  Proofs of Main Text

## E.1  Theorem 2.1

*Proof of Theorem 2.1.*

$$\underset{\mathbf{p}\in\Delta^d}{\text{Max}}\left[\langle\mathbf{p},\mathbf{z}\rangle-\frac{1}{2}\|\mathbf{p}\|^2+\frac{1}{2}\right]=\underset{\mathbf{p}\in\Delta^d}{\text{Max}}\left[\frac{1}{2}\|\mathbf{z}\|^2+\langle\mathbf{p},\mathbf{z}\rangle-\frac{1}{2}\|\mathbf{p}\|^2-\frac{1}{2}\|\mathbf{z}\|^2+\frac{1}{2}\right] \tag{E.1}$$

$$=\underset{\mathbf{p}\in\Delta^d}{\text{Max}}\left[\frac{1}{2}\|\mathbf{z}\|^2+\frac{1}{2}-\frac{1}{2}\|\mathbf{p}-\mathbf{z}\|^2\right] \tag{E.2}$$

$$=\frac{1}{2}\|\mathbf{z}\|^2+\frac{1}{2}-\underset{\mathbf{p}\in\Delta^d}{\text{Min}}\left[\frac{1}{2}\|\mathbf{p}-\mathbf{z}\|^2\right] \tag{E.3}$$

$$=\frac{1}{2}\|\mathbf{z}\|^2-\frac{1}{2}\|\mathbf{p}^\star-\mathbf{z}\|^2+\frac{1}{2}=\Psi^\star(\mathbf{z}), \tag{E.4}$$

with $\mathbf{p}^\star$ given by (2.3).  $\square$

## E.2  Lemma 2.1

*Proof of Lemma 2.1.* To show monotonic decreasing property of the energy (2.1), we first derive the sparse retrieval dynamics by utilizing the aforementioned Theorem 2.1, Corollary 2.1.1, along with the convex-concave procedure [Yuille and Rangarajan, 2003, 2001]. Then, we show the monotonicity of $\mathcal{H}$ by constructing an iterative upper bound of $\mathcal{H}$ which is convex in $\mathbf{x}_{t+1}$ and thus, lowered iteratively by the CCCP method.

By convex conjugate, $\Psi^*$, the conjugate convex of $\Psi$, is always convex, and hence $-\Psi^*$ is a concave function. Therefore, the energy function $\mathcal{H}$ is by construction the sum of the convex function $\mathcal{H}_1(\mathbf{x})\coloneqq\frac{1}{2}\langle\mathbf{x},\mathbf{x}\rangle$ and the concave function $\mathcal{H}_2(\mathbf{x})\coloneqq-\Psi^\star(\mathbf{\Xi}^\mathsf{T}\mathbf{x})$. In addition, $\mathcal{H}$ is differentiable by definition.

Applying the convex-concave procedure to $\mathcal{H}$ gives

$$\boldsymbol{\nabla}_{\mathbf{x}}\mathcal{H}_1(\mathbf{x}_{t+1})=-\boldsymbol{\nabla}_{\mathbf{x}}\mathcal{H}_2(\mathbf{x}_t), \tag{E.5}$$

which leads to

$$\mathbf{x}_{t+1}=\boldsymbol{\nabla}_{\mathbf{x}}\Psi(\mathbf{\Xi}\mathbf{x}_t)=\mathbf{\Xi}\,\text{Sparsemax}\left(\mathbf{\Xi}^\mathsf{T}\mathbf{x}_t\right), \tag{E.6}$$

by Theorem 2.1 and Corollary 2.1.1.

Following [Yuille and Rangarajan, 2003, 2001], we show the monotonic decreasing of (2.1) over $t$ with by considering the problem of energy minimization:

$$\underset{\mathbf{x}}{\text{Min}}\left[\mathcal{H}(\mathbf{x})\right]\quad=\quad\underset{\mathbf{x}}{\text{Min}}\left[\mathcal{H}_1(\mathbf{x})+\mathcal{H}_2(\mathbf{x})\right], \tag{E.7}$$

which, in the convex-concave procedure, is solved by iteratively computing

$$\mathbf{x}_{t+1}\quad\in\quad\underset{\mathbf{x}}{\text{ArgMin}}\left[\mathcal{H}_1(\mathbf{x})+\langle\mathbf{x},\boldsymbol{\nabla}_{\mathbf{x}}\mathcal{H}_2(\mathbf{x}_t)\rangle\right], \tag{E.8}$$

for all $t$. The intuition behind this is to linearize the concave $\mathcal{H}_2$ around the current iteration's solution $\mathbf{x}_t$, making $\mathcal{H}_1(\mathbf{x}_{t+1})+\langle\mathbf{x}_{t+1},\boldsymbol{\nabla}_{\mathbf{x}}\mathcal{H}_2(\mathbf{x}_t)\rangle$ convex in $\mathbf{x}_{t+1}$.

By convexity and concavity of $\mathcal{H}_1$ and $\mathcal{H}_2$, we have

$$\mathcal{H}_1(\mathbf{x})\geq\mathcal{H}_1(\mathbf{y})+\langle(\mathbf{x}-\mathbf{y}),\boldsymbol{\nabla}_{\mathbf{x}}\mathcal{H}_1(\mathbf{y})\rangle, \tag{E.9}$$

$$\mathcal{H}_2(\mathbf{x})\leq\mathcal{H}_2(\mathbf{y})+\langle(\mathbf{x}-\mathbf{y}),\boldsymbol{\nabla}_{\mathbf{x}}\mathcal{H}_2(\mathbf{y})\rangle, \tag{E.10}$$

for all $\mathbf{x},\mathbf{y}$. Therefore, it holds

$$\mathcal{H}(\mathbf{x})=\mathcal{H}_1(\mathbf{x})+\mathcal{H}_2(\mathbf{x}) \tag{E.11}$$

$$\leq\mathcal{H}_1(\mathbf{x})+\mathcal{H}_2(\mathbf{y})+\langle(\mathbf{x}-\mathbf{y}),\boldsymbol{\nabla}_{\mathbf{x}}\mathcal{H}_2(\mathbf{y})\rangle\coloneqq\mathcal{H}_U(\mathbf{x},\mathbf{y}), \tag{E.12}$$

where $\mathcal{H}_U$ is the upper bound of $\mathcal{H}$. Then, for each iteration $t$, we have

$$\mathbf{x}_{t+1}\in\underset{\mathbf{x}}{\text{ArgMin}}\left[\mathcal{H}_U(\mathbf{x},\mathbf{x}_t)\right]=\underset{\mathbf{x}}{\text{ArgMin}}\left[\mathcal{H}_1(\mathbf{x})+\langle\mathbf{x},\boldsymbol{\nabla}_{\mathbf{x}}\mathcal{H}_2(\mathbf{x}_t)\rangle\right], \tag{E.13}$$

which lowers the upper bound $\mathcal{H}_U$ iteratively and hence decreases the value of $\mathcal{H}$ monotonically, i.e.

$$
\begin{aligned}
\mathcal{H}(\mathbf{x}_{t+1}) &\leq \mathcal{H}_U(\mathbf{x}_{t+1}, \mathbf{x}_t) && \text{(By (E.12))} \\
&\leq \mathcal{H}_U(\mathbf{x}_t, \mathbf{x}_t) && \text{(Set } \mathbf{x} = \mathbf{y} \text{ in (E.12))} \\
&= \mathcal{H}(\mathbf{x}_t), && \text{(E.14)}
\end{aligned}
$$

for all $t$. This completes the proof that $\mathcal{H}$ can be monotonically decreased by $\mathcal{T}(\mathbf{x})$ given by (2.5). $\quad\square$

### E.3 Theorem 2.2

*Proof of Theorem 2.2.* Let $\mathcal{T}_{\text{Dense}}$ be the retrieval dynamics of the dense modern Hopfield model [Ramsauer et al., 2021], and $\|\mathcal{T}(\mathbf{x}) - \boldsymbol{\xi}_\mu\|$ and $\|\mathcal{T}_{\text{Dense}}(\mathbf{x}) - \boldsymbol{\xi}_\mu\|$ be the retrieval error of sparse and dense Hopfield model, respectively.

We observe

$$
\|\mathcal{T}(\mathbf{x}) - \boldsymbol{\xi}_\mu\| - \|\mathcal{T}_{\text{Dense}}(\mathbf{x}) - \boldsymbol{\xi}_\mu\|
$$

$$
= \left\| \sum_{\nu=1}^{\kappa} \boldsymbol{\xi}_\nu \left[\text{Sparsemax}\left(\beta \boldsymbol{\Xi}^\mathsf{T} \mathbf{x}\right)\right]_\nu - \boldsymbol{\xi}_\mu \right\| - \left\| \sum_{\nu=1}^{\kappa} \boldsymbol{\xi}_\nu \left[\text{Softmax}\left(\beta \boldsymbol{\Xi}^\mathsf{T} \mathbf{x}\right)\right]_\nu - \boldsymbol{\xi}_\mu \right\| \tag{E.15}
$$

$$
\leq \left\| \sum_{\nu=1}^{\kappa} \left[\text{Sparsemax}(\beta \boldsymbol{\Xi}^\mathsf{T} \mathbf{x})\right]_\nu \boldsymbol{\xi}_\nu \right\| - \left\| \sum_{\nu=1}^{\kappa} \left[\text{Softmax}\left(\beta \boldsymbol{\Xi}^\mathsf{T} \mathbf{x}\right)\right]_\nu \boldsymbol{\xi}_\nu \right\| \tag{E.16}
$$

$$
\leq 0, \tag{E.17}
$$

which gives

$$
\|\mathcal{T}(\mathbf{x}) - \boldsymbol{\xi}_\mu\| \quad \leq \quad \|\mathcal{T}_{\text{Dense}}(\mathbf{x}) - \boldsymbol{\xi}_\mu\|. \tag{E.18}
$$

Next, we provide an upper bound of the sparse retrieval error for a query $\mathbf{x} \in S_\mu$ given memory patterns $\{\boldsymbol{\xi}_\nu\}_{\nu \in [M]}$.

According to the (2.3), it holds

$$
[\text{Sparsemax}\left(\beta \boldsymbol{\Xi}^\mathsf{T} \mathbf{x}\right)]_\mu \leq \left[\beta \boldsymbol{\Xi}^\mathsf{T} \mathbf{x}\right]_\mu - \left[\beta \boldsymbol{\Xi}^\mathsf{T} \mathbf{x}\right]_{(\kappa)} + \frac{1}{\kappa}, \tag{E.19}
$$

for all $\mu \in [M]$. Then, the sparse retrieval error is

$$
\|\mathcal{T}(\mathbf{x}) - \boldsymbol{\xi}^\mu\| = \left\| \boldsymbol{\Xi} \,\text{Sparsemax}\left(\beta \boldsymbol{\Xi}^\mathsf{T} \mathbf{x}\right) - \boldsymbol{\xi}^\mu \right\| = \left\| \sum_{\nu=1}^{\kappa} \boldsymbol{\xi}_{(\nu)} \left[\text{Sparsemax}\left(\beta \boldsymbol{\Xi}^\mathsf{T} \mathbf{x}\right)\right]_{(\nu)} - \boldsymbol{\xi}^\mu \right\|
$$

$$
\leq m + m\beta \left\| \sum_{\nu=1}^{\kappa} \left( \left[\boldsymbol{\Xi}^\mathsf{T} \mathbf{x}\right]_{(\nu)} - \left[\boldsymbol{\Xi}^\mathsf{T} \mathbf{x}\right]_{(\kappa)} + \frac{1}{\beta\kappa} \right) \frac{\boldsymbol{\xi}_{(\nu)}}{m} \right\| \qquad \text{(By (E.19))} \tag{E.20}
$$

$$
= m + d^{1/2} m\beta \left[ \sum_{\nu=1}^{\kappa} \left( \left[\boldsymbol{\Xi}^\mathsf{T} \mathbf{x}\right]_{(\nu)} - \left[\boldsymbol{\Xi}^\mathsf{T} \mathbf{x}\right]_{(\kappa)} + \frac{1}{\beta\kappa} \right) \right] \tag{E.20}
$$

$$
\leq m + d^{1/2} m\beta \left[ \kappa \left( \underset{\nu \in [M]}{\text{Max}} \langle \boldsymbol{\xi}_\nu, \mathbf{x} \rangle - \left[\boldsymbol{\Xi}^\mathsf{T} \mathbf{x}\right]_{(\kappa)} \right) + \frac{1}{\beta} \right]. \tag{E.21}
$$

$\square$

### E.4 Lemma 2.2

In order to prove Lemma 2.2, we need the following two auxiliary lemmas.

**Lemma E.1** ([Gunawardana et al., 2005], Proposition 7). Let $\mathbf{x}_t \in \mathcal{X}_t$ and $\mathbf{x}_{t+1} \in \mathcal{X}_{t+1}$. Given a real-valued continuous function $\mathcal{H}_U$ on $\mathcal{X}_t \times \mathcal{X}_{t+1}$, define the point-to-set map $\mathcal{T} : \mathcal{X}_t \to \mathcal{X}_{t+1}$ by

$$
\mathcal{T}(\mathbf{x}_t) := \underset{\mathbf{x}'_{t+1} \in \mathcal{X}_{t+1}}{\text{ArgMin}} \mathcal{H}_U(\mathbf{x}_t, \mathbf{x}'_{t+1}) \tag{E.22}
$$

$$
= \{\mathbf{x}_{t+1} \mid \mathcal{H}_U(\mathbf{x}_t, \mathbf{x}_{t+1}) \leq \mathcal{H}_U(\mathbf{x}_t, \mathbf{x}'_{t+1}), \forall \mathbf{x}'_{t+1} \in \mathcal{X}_{t+1}\}. \tag{E.23}
$$

Then $\mathcal{T}$ is a closed map at $\mathbf{x}_t$ if $\mathcal{T}(\mathbf{x})$ is non-empty.

**Lemma E.2** ([Sriperumbudur and Lanckriet, 2009], Lemma 5). Recall a fixed point of $\mathcal{T}$ w.r.t. $\mathcal{H}$ is a point for which $\mathbf{x} = \mathcal{T}(\mathbf{x})$, and a generalized fixed point is a point for which $\mathbf{x} \in \mathcal{T}(\mathbf{x})$. Suppose $\mathbf{x}^{\star}$ is a generalized fixed point of $\mathcal{T}$, then, $\mathbf{x}^{\star}$ is a stationary point of the minimization problem (E.7).

*Proof of Lemma 2.2.* From Zangwill global convergence theory for iterative algorithms [Zangwill, 1969], all limit points of $\{\mathbf{x}_t\}_{t=0}^{\infty}$ are generalized fixed points[10], if the energy function $\mathcal{H}$ and the retrieval dynamics $\mathcal{T}$ satisfy the following three conditions.

(i) For any sequence $\{\mathbf{x}_t\}_{t=0}^{\infty}$ with starting point $\mathbf{x}_0 \in S_{\mu}$, all points in this sequence are in the same compact set $S_{\mu}$.

(ii) $\mathcal{H}$ is monotonically decreased by $\mathcal{T}(\mathbf{x})$, i.e. $\mathcal{H}(\mathbf{x}_{t+1}) \leq \mathcal{H}(\mathbf{x}_t), \forall \mathbf{x}_{t+1} = \mathcal{T}(\mathbf{x}_t)$.

(iii) For all $t$, if $\mathcal{H}(\mathbf{x}_{t+1}) < \mathcal{H}(\mathbf{x}_t)$, $\mathcal{T}$ is closed at $\mathbf{x}_t$.

From Definition 2.2, since $S_{\mu}$ with finite radius $R$ is bounded and closed, every $S_{\mu}$ is a compact set. Namely, for any sequence $\{\mathbf{x}_t\}_{t=0}^{\infty}$, all points are embedded in the compact set $S_{\mu}$. Therefore, condition (i) is automatically satisfied. Then condition (ii), the monotonic descent property of $\{\mathbf{x}_t\}_{t=0}^{\infty}$, has been analyzed in the original paper of CCCP [Yuille and Rangarajan, 2003]. By our definition on $\mathcal{H}_1$ and $\mathcal{H}_2$, we have $\mathcal{H}_U(\mathbf{x}, \mathbf{y}) := \mathcal{H}_1(\mathbf{x}) + \mathcal{H}_2(\mathbf{y}) + \langle (\mathbf{x} - \mathbf{y}), \boldsymbol{\nabla}_{\mathbf{x}} \mathcal{H}_2(\mathbf{y}) \rangle$ is continuous in $\mathbf{x}$ and $\mathbf{y}$. Consequently, by Lemma E.1, condition (iii) holds due to the non-empty assumption on the point-to-set map $\mathcal{T}$. Thus, by Zangwill global convergence theory, all the limit points of $\{\mathbf{x}_t\}_{t=0}^{\infty}$ are fixed points of $\mathcal{T}$. Subsequently, by the results of Lemma E.2, these fixed points are also the stationary points of the minimization problem (E.7). Therefore, the energy function is ensured to converge to local optimum. □

---

[10]Recall that, a generalized fixed point of $\mathcal{T}$ is defined as $\mathbf{x}^{\star} := \{\mathbf{x} \mid \mathbf{x} \in \mathcal{T}(\mathbf{x})\}$.

## E.5 Theorem 3.1 and Corollary 3.1.1

*Proof of Theorem 3.1.* Recall $n := \|\mathbf{x}\|$. By Definition 3.1, we have

$$\underset{\mu \in [M]}{\text{Max}} \langle \boldsymbol{\xi}_\mu, \mathbf{x} \rangle = \langle \boldsymbol{\xi}_\nu, \mathbf{x} \rangle - \widetilde{\Delta}_\nu, \tag{E.24}$$

thereby obtaining

$$\|\mathcal{T}(\mathbf{x}) - \boldsymbol{\xi}_\mu\| \quad \leq \quad m + d^{1/2} m \beta \left[ \kappa \left( \underset{\nu \in [M]}{\text{Max}} \langle \boldsymbol{\xi}_\nu, \mathbf{x} \rangle - \left[ \boldsymbol{\Xi}^{\mathsf{T}} \mathbf{x} \right]_{(\kappa)} \right) + \frac{1}{\beta} \right] \tag{E.25}$$

$$= \quad m + d^{1/2} m \beta \left[ \kappa \left( \langle \boldsymbol{\xi}_\mu, \mathbf{x} \rangle - \widetilde{\Delta}_\mu - \left[ \boldsymbol{\Xi}^{\mathsf{T}} \mathbf{x} \right]_{(\kappa)} \right) + \frac{1}{\beta} \right] \tag{E.26}$$

Since $n := \|\mathbf{x}\|$ and $m := \max_\mu \|\boldsymbol{\xi}^\mu\|$, we have

$$m + d^{1/2} m \beta \left[ \kappa \left( \langle \boldsymbol{\xi}_\mu, \mathbf{x} \rangle - \widetilde{\Delta}_\mu - \left[ \boldsymbol{\Xi}^{\mathsf{T}} \mathbf{x} \right]_{(\kappa)} \right) + \frac{1}{\beta} \right] \tag{E.27}$$

$$\leq m + d^{1/2} m \beta \left[ \kappa \left( mn - \widetilde{\Delta}_\mu - \left[ \boldsymbol{\Xi}^{\mathsf{T}} \mathbf{x} \right]_{(\kappa)} \right) + \frac{1}{\beta} \right] \tag{E.28}$$

Then, by the Cauchy-Schwartz inequality

$$|\langle \boldsymbol{\xi}_\mu, \boldsymbol{\xi}_\mu \rangle - \langle \mathbf{x}, \boldsymbol{\xi}_\mu \rangle| \leq \|\boldsymbol{\xi}_\mu - \mathbf{x}\| \cdot \|\boldsymbol{\xi}_\mu\| \leq \|\boldsymbol{\xi}_\mu - \mathbf{x}\| m, \quad \forall \mu \in [M], \tag{E.29}$$

we observe that $\widetilde{\Delta}_\mu$ can be expressed in terms of $\Delta_\mu$:

$$\widetilde{\Delta}_\mu = \underset{\nu, \nu \neq \mu}{\text{Min}} \left[ \langle \mathbf{x}, \boldsymbol{\xi}_\mu \rangle - \langle \mathbf{x}, \boldsymbol{\xi}_\nu \rangle + (\langle \boldsymbol{\xi}_\mu, \boldsymbol{\xi}_\mu \rangle - \langle \boldsymbol{\xi}_\mu, \boldsymbol{\xi}_\nu \rangle) - (\langle \boldsymbol{\xi}_\mu, \boldsymbol{\xi}_\mu \rangle - \langle \boldsymbol{\xi}_\mu, \boldsymbol{\xi}_\nu \rangle) \right] \tag{E.30}$$

$$\geq \underset{\nu, \nu \neq \mu}{\text{Min}} \left[ \langle \boldsymbol{\xi}_\mu, \boldsymbol{\xi}_\mu \rangle - \langle \boldsymbol{\xi}_\mu, \boldsymbol{\xi}_\nu \rangle + (\langle \boldsymbol{\xi}_\mu, \boldsymbol{\xi}_\nu \rangle - \langle \mathbf{x}, \boldsymbol{\xi}_\nu \rangle) - (\langle \boldsymbol{\xi}_\mu, \boldsymbol{\xi}_\mu \rangle - \langle \mathbf{x}, \boldsymbol{\xi}_\mu \rangle) \right]$$
$$\text{(By Cauchy-Schwarz)}$$

$$= \Delta_\mu - 2\|\boldsymbol{\xi}_\mu - \mathbf{x}\| m = \Delta_\mu - 2mR, \qquad\qquad \text{(By } \mathbf{x} \in S_\mu\text{)}$$

where $R$ is radius of the sphere $S_\mu$. Inserting the bound on $\widetilde{\Delta}_\mu$, we obtain

$$\|\mathcal{T}(\mathbf{x}) - \boldsymbol{\xi}_\mu\| \leq m + d^{1/2} m \beta \left[ \kappa \left( mn - \Delta_\mu + 2mR - \left[ \boldsymbol{\Xi}^{\mathsf{T}} \mathbf{x} \right]_{(\kappa)} \right) + \frac{1}{\beta} \right]. \tag{E.31}$$

For $\mathcal{T}$ to be a mapping from $S_\mu$ to $S_\mu$, we obtain the inequality:

$$m + d^{1/2} m \beta \left[ \kappa \left( mn - \Delta_\mu + 2mR - \left[ \boldsymbol{\Xi}^{\mathsf{T}} \mathbf{x} \right]_{(\kappa)} \right) + \frac{1}{\beta} \right] \leq R, \tag{E.32}$$

which gives

$$\Delta_\mu \geq mn + 2mR - \left[ \boldsymbol{\Xi}^{\mathsf{T}} \mathbf{x} \right]_{(\kappa)} - \frac{1}{\kappa} \left( \frac{R - m - md^{1/2}}{m \beta d^{1/2}} \right). \tag{E.33}$$

Therefore, as long as $\Delta_\mu$ satisfies this inequality, $\mathcal{T}$ is a mapping from $S_\mu$ onto itself. $\qquad\square$

*Proof of Corollary 3.1.1.* Let $\mathcal{T}_{\text{Dense}}$ be the retrieval dynamics of the dense modern Hopfield model [Ramsauer et al., 2021], and $\epsilon_{\text{Sparsemax}} := \|\mathcal{T}(\mathbf{x}) - \boldsymbol{\xi}_\mu\|$ and $\epsilon_{\text{Dense}} := \|\mathcal{T}_{\text{Dense}}(\mathbf{x}) - \boldsymbol{\xi}_\mu\|$ be the retrieval error of sparse and dense Hopfield model, respectively.

First, let's recall Theorem 2.2, which states that

$$\epsilon_{\text{Sparsemax}} = \|\mathcal{T}(\mathbf{x}) - \boldsymbol{\xi}_\mu\| \quad \leq \quad \epsilon_{\text{Dense}} = \|\mathcal{T}_{\text{Dense}}(\mathbf{x}) - \boldsymbol{\xi}_\mu\|. \tag{E.34}$$

Next we want to find the lower bound of separation $\Delta_\mu$ such that $\mathcal{T}$ is a mapping from $S_\mu$ onto $S_\mu$.

To link $\Delta_\mu$ to $\mathcal{T}$, we first bound $\epsilon_{\text{Dense}}$ via [Ramsauer et al., 2021, Lemma A.4]:

$$\epsilon_{\text{Dense}} = \|\mathcal{T}_{\text{Dense}}(\mathbf{x}) - \boldsymbol{\xi}_\mu\| \tag{E.35}$$

$$= \left\| \boldsymbol{\xi}_\mu - \sum_{\nu=1}^{M} [\text{Softmax}(\beta \Xi^\mathsf{T} \mathbf{x})]_\nu \boldsymbol{\xi}_\nu \right\| \tag{E.36}$$

$$= \left\| \left(1 - \left[\text{Softmax}(\beta \Xi^\mathsf{T} \mathbf{x})\right]_\mu\right) \boldsymbol{\xi}_\mu - \sum_{\nu=1,\nu\neq\mu}^{M} \left[\text{Softmax}(\beta \Xi^\mathsf{T} \mathbf{x})\right]_\nu \boldsymbol{\xi}_\nu \right\| \tag{E.37}$$

$$\leq \widetilde{\epsilon} \|\boldsymbol{\xi}_\mu\| + \frac{\widetilde{\epsilon}}{M-1} \sum_{\nu=1,\nu\neq\mu}^{M} \|\boldsymbol{\xi}_\nu\| \tag{E.38}$$

$$\leq \widetilde{\epsilon} \left( m + \frac{1}{M-1} \sum_{\nu=1,\nu\neq\mu}^{M} m \right) \tag{E.39}$$

$$\leq 2\widetilde{\epsilon} m, \tag{E.40}$$

where $\widetilde{\epsilon} := (M-1)\exp\left\{-\beta\widetilde{\Delta}_\mu\right\} = (M-1)\exp\left\{-\beta\left(\langle\boldsymbol{\xi}_\mu,\mathbf{x}\rangle - \text{Max}_{\nu\in[M],\nu\neq\mu}\langle\boldsymbol{\xi}_\mu,\boldsymbol{\xi}_\nu\rangle\right)\right\}$ and the inequality

$$[\text{Softmax}(\beta \Xi^\mathsf{T} \mathbf{x})]_\nu = \frac{\exp\{\beta\left(\langle\mathbf{x},\boldsymbol{\xi}_\nu\rangle - \langle\mathbf{x},\boldsymbol{\xi}_\mu\rangle\right)\}}{1 + \sum_{\nu'\neq\mu}\exp\{\beta\left(\langle\mathbf{x},\boldsymbol{\xi}_{\nu'}\rangle - \langle\mathbf{x},\boldsymbol{\xi}_\mu\rangle\right)\}} \leq \exp\left\{-\beta\widetilde{\Delta}_\mu\right\}, \tag{E.41}$$

is used in the fourth line.

Then, by the Cauchy-Schwartz inequality

$$|\langle\boldsymbol{\xi}_\mu,\boldsymbol{\xi}_\mu\rangle - \langle\mathbf{x},\boldsymbol{\xi}_\mu\rangle| \leq \|\boldsymbol{\xi}_\mu - \mathbf{x}\| \cdot \|\boldsymbol{\xi}_\mu\| \leq \|\boldsymbol{\xi}_\mu - \mathbf{x}\| m, \quad \forall \mu \in [M], \tag{E.42}$$

we observe that $\widetilde{\Delta}_\mu$ can be expressed in terms of $\Delta_\mu$:

$$\widetilde{\Delta}_\mu = \min_{\nu,\nu\neq\mu} \left[\langle\mathbf{x},\boldsymbol{\xi}_\mu\rangle - \langle\mathbf{x},\boldsymbol{\xi}_\nu\rangle + (\langle\boldsymbol{\xi}_\mu,\boldsymbol{\xi}_\mu\rangle - \langle\boldsymbol{\xi}_\mu,\boldsymbol{\xi}_\nu\rangle) - (\langle\boldsymbol{\xi}_\mu,\boldsymbol{\xi}_\mu\rangle - \langle\boldsymbol{\xi}_\mu,\boldsymbol{\xi}_\nu\rangle)\right] \tag{E.43}$$

$$\geq \min_{\nu,\nu\neq\mu} \left[\langle\boldsymbol{\xi}_\mu,\boldsymbol{\xi}_\mu\rangle - \langle\boldsymbol{\xi}_\mu,\boldsymbol{\xi}_\nu\rangle + (\langle\boldsymbol{\xi}_\mu,\boldsymbol{\xi}_\nu\rangle - \langle\mathbf{x},\boldsymbol{\xi}_\nu\rangle) - (\langle\boldsymbol{\xi}_\mu,\boldsymbol{\xi}_\mu\rangle - \langle\mathbf{x},\boldsymbol{\xi}_\mu\rangle)\right]$$
$$\text{(By Cauchy-Schwarz)}$$

$$= \Delta_\mu - 2\|\boldsymbol{\xi}_\mu - \mathbf{x}\| m = \Delta_\mu - 2mR, \quad\quad\quad\quad\quad\quad \text{(By } \mathbf{x} \in S_\mu\text{)}$$

where $R$ is radius of the sphere $S_\mu$.

Hence, combining the bound from (E.40) with (E.34) results in

$$\|\mathcal{T}(\mathbf{x}) - \boldsymbol{\xi}_\mu\| \leq \|\mathcal{T}_{\text{Dense}}(\mathbf{x}) - \boldsymbol{\xi}_\mu\| \leq 2\widetilde{\epsilon} m \tag{E.44}$$

$$= 2(M-1)\exp\left\{-\beta\widetilde{\Delta}_\mu\right\} m \tag{E.45}$$

$$\leq 2(M-1)\exp\{-\beta\left(\Delta_\mu - 2mR\right)\} m. \tag{E.46}$$

Therefore, given $\delta := \|\mathcal{T}_{\text{Dense}}(\mathbf{x}) - \boldsymbol{\xi}_\mu\| - \|\mathcal{T}(\mathbf{x}) - \boldsymbol{\xi}_\mu\| \leq 0$, we have

$$\|\mathcal{T}(\mathbf{x}) - \boldsymbol{\xi}_\mu\| \leq 2(M-1)\exp\{-\beta\left(\Delta_\mu - 2mR + \delta\right)\} m - \delta \leq \|\mathcal{T}_{\text{Dense}}(\mathbf{x}) - \boldsymbol{\xi}_\mu\|. \tag{E.47}$$

For $\mathcal{T}$ to be a mapping from $S_\mu$ onto $S_\mu$, it is sufficient to have

$$2(M-1)\exp\{-\beta(\Delta_\mu - 2mR)\} m - \delta \leq R, \tag{E.48}$$

which leads to

$$\Delta_\mu \geq \frac{1}{\beta} \ln\left(\frac{2(M-1)m}{R + \delta}\right) + 2mR. \tag{E.49}$$

$\square$

## E.6   Lemma 3.1

We begin with a helper lemma.

**Lemma E.3** ([Ramsauer et al., 2021]). Given real numbers $a, b \in \mathbb{R}$. If the equation

$$ac + c \ln c - b = 0, \tag{E.50}$$

holds, then the solution is

$$c = \frac{b}{W_0(\exp(a + \ln b))}. \tag{E.51}$$

*Proof.* Starting from the given equation, we can rearrange and solve for $c$ as follows:

$$ac + c \ln c - b = 0,$$
$$a + \ln c = \frac{b}{c},$$
$$\frac{b}{c} + \ln\left(\frac{b}{c}\right) = a + \ln b,$$
$$\frac{b}{c} \exp\left(\frac{b}{c}\right) = \exp(a + \ln b),$$
$$\frac{b}{c} = W_0(\exp(a + \ln b)),$$
$$c = \frac{b}{W_0(\exp(a + \ln b))}.$$

This completes the proof. $\qquad\square$

Then we present the proof.

*Proof of Lemma 3.1.* Equipped with $\Delta_\mu \geq \frac{1}{\beta} \ln\left(\frac{2(M-1)m}{R+\delta}\right) + 2mR$ from Corollary 3.1.1, we first write down the probability of success storage and retrieval, i.e. minimal separation $\Delta_{\min}$ satisfies well-separation condition.

Let $\Delta_{\min} = \text{Min}_{\mu \in [M]} \Delta_\mu$, and $\theta_{\mu\nu}$ be the angle between two patterns $\boldsymbol{\xi}^\mu$ and $\boldsymbol{\xi}^\nu$. Intuitively, $\theta_{\mu\nu} \in [0, \pi]$ represent the pairwise correlation of two patterns the two patterns.

We have

$$\Delta_{\min} \geq \frac{1}{\beta} \ln\left(\frac{2(M-1)m}{R+\delta}\right) + 2mR, \tag{E.52}$$

and

$$\Delta_{\min} = \underset{1 \leq \mu \leq \nu \leq M}{\text{Min}} \left[m^2 \left(1 - \cos(\theta_{\mu\nu})\right)\right] = m^2 \left[1 - \cos(\theta_{\min})\right], \tag{E.53}$$

where $\theta_{\min} := \text{Min}_{1 \leq \mu \leq \nu \leq M} \theta_{\mu\nu} \in [0, \pi]$. Then, it holds

$$m^2 \left[1 - \cos(\theta_{\min})\right] \geq \frac{1}{\beta} \ln\left(\frac{2(M-1)m}{R+\delta}\right) + 2mR. \tag{E.54}$$

With Corollary 3.1.1 , we write down the probability of success storage and retrieval as

$$P\left(\Delta_\mu \geq \frac{1}{\beta} \ln\left(\frac{2(M-1)m}{R+\delta}\right) + 2mR\right) = 1 - p. \tag{E.55}$$

By (E.54), we have

$$P\left(m^2 \left[1 - \cos(\theta_{\min})\right] \geq \frac{1}{\beta} \ln\left(\frac{2(M-1)m}{R+\delta}\right) + 2mR\right) = 1 - p. \tag{E.56}$$

By [Olver et al., 2010, (4.22.2)], for $0 \leq \cos(\theta_{\min}) \leq 1$, $\cos(\theta_{\min})$ can be upper bounded by:

$$\cos(\theta_{\min}) \leq 1 - \frac{\theta_{\min}^2}{5}. \tag{E.57}$$

It holds

$$P\left(\frac{m^2 \theta_{\min}^2}{5} \geq \frac{1}{\beta} \ln\left(\frac{2(M-1)m}{R+\delta}\right) + 2mR\right) = 1 - p, \tag{E.58}$$

which can be rewritten as

$$P\left(M^{\frac{2}{d-1}}\theta_{\min} \geq \frac{\sqrt{5}M^{\frac{2}{d-1}}}{m}\left[\frac{1}{\beta}\ln\left(\frac{2(M-1)m}{R+\delta}\right) + 2mR\right]^{\frac{1}{2}}\right) = 1 - p. \tag{E.59}$$

Here, $M^{2/d-1}$ is introduced for later convenience.

Let

$$\omega_d := \frac{2\pi^{d+1/2}}{\Gamma\left(\frac{d+1}{2}\right)}, \tag{E.60}$$

be the surface area of a $d$-dimensional unit sphere is, where $\Gamma(\cdot)$ represents the gamma function.

By [Brauchart et al., 2018, Lemma 3.5], we obtain

$$P\left(M^{\frac{2}{d-1}}\theta_{\min} \geq \frac{\sqrt{5}M^{\frac{2}{d-1}}}{m}\left[\frac{1}{\beta}\ln\left(\frac{2(M-1)m}{R+\delta}\right) + 2mR\right]^{\frac{1}{2}}\right) = 1 - p$$

$$\geq 1 - \frac{1}{2}\gamma_{d-1}5^{\frac{d-1}{2}}M^2 m^{-(d-1)}\left[\frac{1}{\beta}\ln\left(\frac{2(M-1)m}{R+\delta}\right) + 2mR\right]^{\frac{d-1}{2}}, \tag{E.61}$$

where $\gamma_d$ is defined as the ratio of surface areas of $(d-1)$- and $d$-dimensional unit sphere:

$$\gamma_d := \frac{1}{d}\frac{\omega_{d-1}}{\omega_d} = \frac{1}{d\sqrt{\pi}}\frac{\Gamma\left(\frac{d+1}{2}\right)}{\Gamma\left(\frac{d}{2}\right)}. \tag{E.62}$$

Recall $d, M \in \mathbb{N}_+$, $p \in [0, 1]$. With some real value $C \in \mathbb{R}$, it holds

$$M = \sqrt{p}C^{\frac{d-1}{4}}. \tag{E.63}$$

From (E.61), we have

$$5^{\frac{d-1}{2}}\left(\sqrt{p}C^{\frac{d-1}{4}}\right)^2 m^{-(d-1)}\left\{\frac{1}{\beta}\ln\left[\frac{2\left(\sqrt{p}C^{\frac{d-1}{4}} - 1\right)m}{R+\delta}\right] + \frac{1}{\beta}\right\}^{\frac{d-1}{2}} - p \leq 0, \tag{E.64}$$

which leads to

$$5^{\frac{d-1}{2}}C^{\frac{d-1}{2}}m^{-(d-1)}\left\{\frac{1}{\beta}\ln\left[\frac{2\left(\sqrt{p}C^{\frac{d-1}{4}} - 1\right)m}{R+\delta}\right] + \frac{1}{\beta}\right\}^{\frac{d-1}{2}} \leq 1. \tag{E.65}$$

To apply Lemma E.3, we first rearrange (E.65) as

$$\frac{5C}{m^2\beta}\left\{\ln\left[\frac{2\left(\sqrt{p}C^{\frac{d-1}{4}} - 1\right)m}{R+\delta}\right] + 1\right\} - 1 \leq 0, \tag{E.66}$$

and then identify

$$a := \frac{4}{d-1}\left\{\ln\left[\frac{2m(\sqrt{p} - 1)}{R+\delta}\right] + 1\right\}, \quad b := \frac{4m^2\beta}{5(d-1)}. \tag{E.67}$$

By Lemma E.3, we have the solution

$$C = \frac{b}{W_0(\exp\{a + \ln b\})}, \tag{E.68}$$

where $W_0(\cdot)$ is the upper branch of the Lambert $W$ function. Since the domain of the Lambert $W$ function is $x > -1/e$ and the fact $\exp\{a + \ln b\} > 0$, the solution exists.

When $C$ satisfies inequality (E.65), we arrive a lower bound on the exponential storage capacity $M$:

$$M \geq \sqrt{p} C^{\frac{d-1}{4}}. \tag{E.69}$$

Notably, the above takes similar form as [Ramsauer et al., 2021, Theorem 3]. To see the blessings of sparsity, we consider the following asymptotic analysis and compare with results from the dense modern Hopfield model. To compare with results from dense modern Hopfield model, we denote the dense counterparts of $a, b$ with $\tilde{\cdot}$ notation, i.e.

$$\tilde{a} := \frac{2}{d-1}\left[1 + \ln\left(2\beta m^2 p\right)\right], \quad \tilde{b} = b. \tag{E.70}$$

By [Corless et al., 1996], for sufficient large $z$, $W_0(z)$ is asymptotic to

$$W_0(z) \simeq \ln z - \ln \ln z + \mathcal{O}(1). \tag{E.71}$$

Therefore, for sufficient large $\beta$, we have

$$W_0\left(\exp\{a + \ln b\}\right) \simeq a + \ln b - \ln\left(a + \ln b\right) + \mathcal{O}(1), \tag{E.72}$$

which is dominated by $a$.

For $a$, we have

$$\tilde{a} \leq a, \tag{E.73}$$

and hence

$$W_0\left(\exp\left\{\tilde{a} + \ln \tilde{b}\right\}\right) \leq W_0\left(\exp\{a + \ln b\}\right). \tag{E.74}$$

Therefore, combining above with (E.68), we have

$$\tilde{C} = \frac{b}{W_0\left(\exp\left\{\tilde{a} + \ln \tilde{b}\right\}\right)} \leq \frac{b}{W_0\left(\exp\{a + \ln b\}\right)} = C, \tag{E.75}$$

which states that the lower bound of the sparse capacity is larger than that of [Ramsauer et al., 2021]

$$M = \sqrt{p} C^{\frac{d-1}{4}} \geq \sqrt{p} \tilde{C}^{\frac{d-1}{4}} = M_{\text{Dense}}. \tag{E.76}$$

$\square$

# F   Auxiliary Theoretical Background

**Remark F.1** (Remark on Definition 2.1). To see the equivalence of the two optimization problems, we observe

$$\begin{aligned}
\operatorname*{ArgMax}_{\mathbf{p} \in \Delta^d}\left[\langle \mathbf{p}, \mathbf{z}\rangle - \Psi(\mathbf{p})\right] &= \operatorname*{ArgMax}_{\mathbf{p} \in \Delta^d}\left[\langle \mathbf{p}, \mathbf{z}\rangle - \frac{1}{2}\|\mathbf{p}\|^2\right] \\
&= \operatorname*{ArgMin}_{\mathbf{p} \in \Delta^d}\left[-\frac{1}{2}\left(\|\mathbf{p}\|^2 + \|\mathbf{z}\|^2 - 2\langle \mathbf{p}, \mathbf{z}\rangle\right)\right] \\
&= \operatorname*{ArgMin}_{\mathbf{p} \in \Delta^d}\left[\frac{1}{2}\|\mathbf{p} - \mathbf{z}\|^2\right],
\end{aligned} \tag{F.1}$$

where the last line is obtained by inserting $\|\mathbf{z}\|^2$ as a constant in (F.1).

# G   Additional Experiments

In order to highlight the benefits of the sparse Hopfield model, particularly under conditions of high data sparsity, we broaden our experimental studies with more models. These models include the `SparseHopfield`, `Hopfield`, the attention mechanism [Vaswani et al., 2017], and a attention-based MIL baseline, the gated-attention mechanism [Ilse et al., 2018].

## G.1   Visualization of Experimental Validation of Theoretical Results

We provide visual demonstrations of Section 4.1 in Figure 4.

## G.2   Bit Pattern MIL

To supplement Section 4.2.1, we conduct further numerical investigations on the same MIL tasks (**bag sparsity** and **bag size**) with `SparseHopfield`, `Hopfield`. In these experiments, we contrast the performance of `SparseHopfield` and `Hopfield` (and also `SparseHopfieldPooling` and `HopfieldPooling`) with the attention mechanism [Vaswani et al., 2017] and the gated-attention mechanism [Ilse et al., 2018]. For the **bag size**, we fix the number of positive pattern in a bag to be 1, and vary bag size from 20 to 300. For the **bag sparsity**, we fix the bag size as 200, and inject from 2 to 100 positive patterns in a positive bag, results in 1 to 50 percent of positive patterns in each positive bag. The results are reported in Table 4. For numerical experiments on synthetic datasets, we do not use hyperparameter search due to the simplicity of both model structure and data.

Table 4: **Top (Bag Size):** Accuracy comparison on bit pattern dataset for sparse and dense Hopfield model. We report the average accuracy over 10 runs. The results suggest that the sparse Hopfield model demonstrates a better performance when facing a bag size increase. **Bottom (Bag Sparsity):** Performance comparison on bit pattern dataset for sparse and dense Hopfield model with varying bag sparsity. We report the average accuracy over 10 runs. The results suggest that the sparse Hopfield model demonstrates a better performance across all sparsity.

| Bag Size | 20 | 50 | 100 | 150 | 200 | 300 |
|---|---|---|---|---|---|---|
| Sparse Hopfield | $98.82 \pm 0.34$ | $99.45 \pm 0.19$ | $97.13 \pm 0.11$ | $95.98 \pm 0.12$ | $94.17 \pm 0.01$ | $90.15 \pm 0.30$ |
| Dense Hopfield | $99.65 \pm 0.70$ | $99.51 \pm 0.87$ | $53.90 \pm 0.00$ | $49.51 \pm 0.02$ | $51.92 \pm 0.12$ | $53.83 \pm 0.12$ |
| Sparse Hopfield Pooling | $\mathbf{99.71 \pm 0.06}$ | $100.0 \pm 0.00$ | $100.0 \pm 0.00$ | $\mathbf{99.76 \pm 0.00}$ | $\mathbf{99.76 \pm 0.00}$ | $\mathbf{99.76 \pm 0.00}$ |
| Dense Hopfield Pooling | $99.68 \pm 0.15$ | $100.0 \pm 0.00$ | $100.0 \pm 0.00$ | $76.44 \pm 0.23$ | $49.13 \pm 0.01$ | $52.88 \pm 0.01$ |
| Attention | $87.01 \pm 0.00$ | $74.51 \pm 0.01$ | $45.19 \pm 0.31$ | $53.75 \pm 0.76$ | $46.63 \pm 0.02$ | $53.36 \pm 0.03$ |
| Gated | $87.88 \pm 0.00$ | $63.44 \pm 0.04$ | $75.38 \pm 0.56$ | $73.45 \pm 0.70$ | $71.05 \pm 0.35$ | $49.61 \pm 1.78$ |

| Bag Sparsity | 1% | 2% | 3% | 5% | 10% | 20% | 40% | 50% |
|---|---|---|---|---|---|---|---|---|
| Sparse Hopfield | $95.62 \pm 0.01$ | $95.98 \pm 0.30$ | $99.68 \pm 0.01$ | $100.0 \pm 0.00$ | $100.0 \pm 0.00$ | $100.0 \pm 0.00$ | $100.0 \pm 0.00$ | $100.0 \pm 0.00$ |
| Dense Hopfield | $51.44 \pm 0.01$ | $57.21 \pm 0.01$ | $75.48 \pm 0.01$ | $99.03 \pm 0.11$ | $99.51 \pm 0.02$ | $100.0 \pm 0.00$ | $100.0 \pm 0.00$ | $100.0 \pm 0.00$ |
| Sparse Hopfield Pooling | $\mathbf{99.76 \pm 0.00}$ | $\mathbf{99.68 \pm 0.00}$ | $100.0 \pm 0.00$ | $100.0 \pm 0.00$ | $100.0 \pm 0.00$ | $100.0 \pm 0.00$ | $100.0 \pm 0.00$ | $100.0 \pm 0.00$ |
| Dense Hopfield Pooling | $49.20 \pm 0.00$ | $85.58 \pm 0.10$ | $100.0 \pm 0.00$ | $100.0 \pm 0.00$ | $99.68 \pm 0.00$ | $100.0 \pm 0.00$ | $100.0 \pm 0.00$ | $100.0 \pm 0.00$ |
| Attention | $74.51 \pm 0.01$ | $78.81 \pm 0.04$ | $96.63 \pm 0.02$ | $100.0 \pm 0.00$ | $99.51 \pm 0.01$ | $100.0 \pm 0.00$ | $100.0 \pm 0.00$ | $100.0 \pm 0.00$ |
| Gated | $78.94 \pm 0.41$ | $95.28 \pm 0.35$ | $98.55 \pm 0.00$ | $99.03 \pm 0.01$ | $100.0 \pm 0.00$ | $100.0 \pm 0.00$ | $100.0 \pm 0.00$ | $100.0 \pm 0.00$ |

## G.3   Convergence Analysis

To supplement Section 4.2.1, we also analyze the convergence behavior of the `SparseHopfield` and `Hopfield` numerically. In Figure 5, we plot the loss and accuracy curve for both models on the bit pattern dataset for the **bag size** tasks mentioned in Section 4.2.1. We include various bag sizes in the plot to examine how the loss curve responds to an increase in bag size (i.e., the number of memory patterns, $M$). The results show that, `SparseHopfield` surpasses the `Hopfield` in nearly all bag sizes. Moreover, for the same bag size, `SparseHopfield` always reaches the minimum validation loss faster than `Hopfield`. This provides empirical support for our theoretical prediction outlined in Theorem 2.2. In conjunction with the findings illustrated in Figure 2, Figure 5 reinforces the benefits of utilizing the sparse Hopfield model. In particular, the evidence verifies the claim in Theorem 2.2, demonstrating that the convergence speed of the sparse and dense Hopfield models shows different dependencies on the bag size $M$ in this experiment.

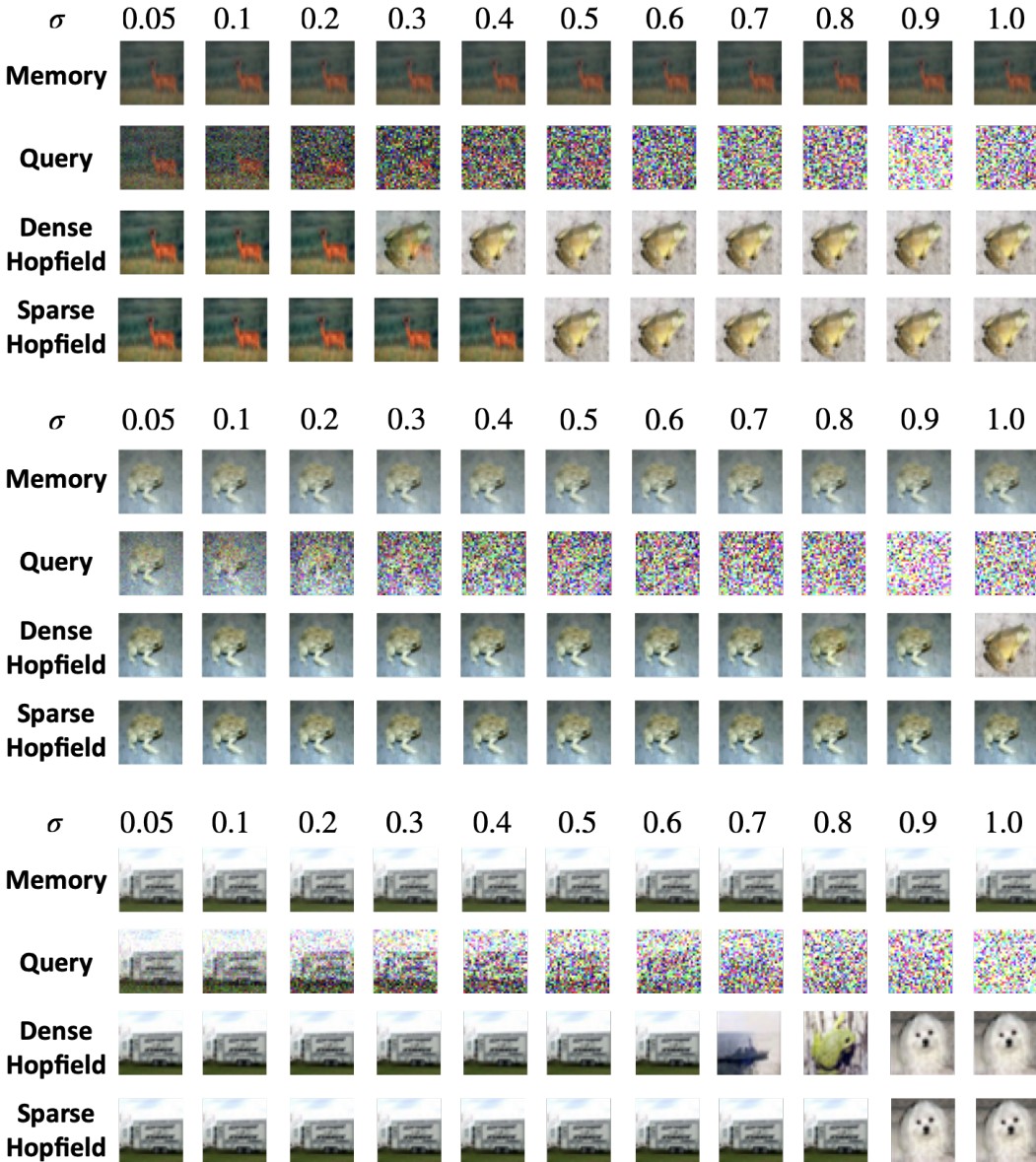

Figure 4: **Visualizing noise-robustness of sparse and dense Hopfield models (Figure 1 of Section 4.1).** We perform memory retrieval using both Dense and Sparse Hopfield models, with queries subjected to varying levels of noise. We randomly select an image from the CIFAR10 dataset to serve as the memory pattern. This selected image is then contaminated with different levels of random noise ($\mu = 0$ and $0.05 \leq \sigma \leq 1.5$) to generate query patterns. The results demonstrate that the Sparse Hopfield model is more effective in retrieving the original image, showcasing its superior robustness against noise.

### G.4  Sparsity Generalization

To supplement Section 4.2.1, we explore another scenario where the bag sparsity shifts between training and test data. We train dense and sparse Hopfield models on a certain bag sparsity, and evaluate on another. The main goal of this setting is to investigate the generalization performance of dense and sparse Hopfield models when the information sparsity shift of training and test data distribution. We fix the bag size to 200, and then implant different number of positive signals to the training and test dataset range from 0.5 to 50 percent. We report the results of `HopfieldPooling`

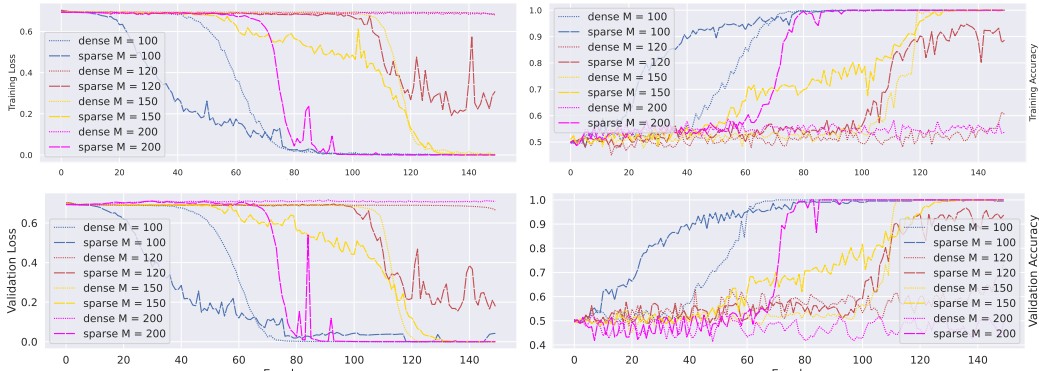

Figure 5: **Top:** The training loss and accuracy curve of `SparseHopfield` and `Hopfield` with different bag sizes. **Bottom:** The validation loss and accuracy curve of `SparseHopfield` and `Hopfield` with different bag sizes. The plotted are the mean of 10 runs. The results indicate that the sparse Hopfield model converges faster than the dense model and also yields superior validation/test accuracy.

and `SparseHopfieldPooling` in Table 5.[11] The result shows that for `HopfieldPooling`, while train on dense bags help its performance of evaluating on sparse bags, lacking ability of learning from sparse bags still affects its performance. Meanwhile `SparseHopfieldPooling` is more robust against sparsity shift, especially for the case where it was trained on dense bags and evaluate on sparse bags. However, both sparse and dense Hopfield models inevitably suffer from a performance drop when having a sparsity gap when train bags are much more sparse than test bags.

Table 5: **Accuracy comparison on bit pattern dataset for sparse and dense Hopfield Model when varying the train/test sparsity gap.** We report the average accuracy over 10 runs. The result shows that for `HopfieldPooling`, while train on dense bags help its performance of evaluating on sparse bags, lacking ability of learning from sparse bags still affects its performance. Meanwhile `SparseHopfieldPooling` is more robust against sparsity shift, especially for the case where it was trained on dense bags and evaluate on sparse bags. However, both sparse and dense Hopfield models inevitably suffer from a performance drop when having a sparsity gap when train bags are much more sparse than test bags.

| # of Test | # of Train Positive Signal per Bag (Dense/Sparse) | | | | | | |
|---|---|---|---|---|---|---|---|
| | 1 | 2 | 10 | 20 | 40 | 80 | 100 |
| 1 | 46.63 / 99.76 | 48.55 / 94.31 | 53.84 / 74.52 | 59.61 / 81.73 | 66.82 / 81.25 | 72.07 / 81.73 | 72.59 / 81.25 |
| 2 | 47.59 / 52.40 | 51.44 / 98.18 | 58.17 / 95.19 | 62.01 / 95.67 | 69.23 / 95.67 | 72.59 / 95.67 | 72.11 / 95.67 |
| 10 | 99.51 / 100.0 | 99.51 / 100.0 | 99.51 / 100.0 | 99.51 / 100.0 | 99.51 / 100.0 | 99.51 / 100.0 | 99.51 / 100.0 |
| 20 | 100.0 / 100.0 | 100.0 / 100.0 | 100.0 / 100.0 | 100.0 / 100.0 | 100.0 / 100.0 | 100.0 / 100.0 | 100.0 / 100.0 |
| 40 | 100.0 / 100.0 | 100.0 / 100.0 | 100.0 / 100.0 | 100.0 / 100.0 | 100.0 / 100.0 | 100.0 / 100.0 | 100.0 / 100.0 |
| 80 | 100.0 / 97.03 | 100.0 / 100.0 | 100.0 / 100.0 | 100.0 / 100.0 | 100.0 / 100.0 | 100.0 / 100.0 | 100.0 / 100.0 |
| 100 | 100.0 / 100.0 | 100.0 / 100.0 | 100.0 / 100.0 | 100.0 / 100.0 | 100.0 / 100.0 | 100.0 / 100.0 | 100.0 / 100.0 |

## G.5   Real-World Experiments

To examine the practical applicability of the proposed model, we implement it in two additional experiments that utilize transformer-based models for distinct tasks. These tasks include multivariate time series prediction [Zhang and Yan, 2022], and neural machine translation [Vaswani et al., 2017]. In these experiments, we substitute the existing attention mechanism with both the `Hopfield` and `SparseHopfield` layers.

---

[11]For the ease of presentation, we exclude the standard deviations in Table 5 as they are all close to zero and less than 0.31%

### G.5.1 Multivariate Time Series Prediction

For the multivariate time series prediction task, we implement two variants of the SOTA Cross-former model [Zhang and Yan, 2022], **Crossformer-DH** and **Crossformer-SH**, with `Hopfield` and `SparseHopfield` layers respectively. These models employ an architecture akin to the Swin-Transformer [Liu et al., 2021], utilizing shifting windows to extract information at multiple resolutions. The experiment results are showed in Table 6. Our results indicate that our proposed `SparseHopfield` not only consistently enhances transformer-based deep learning models but also achieves SOTA performance. In 60+% of 58 settings, the Sparse Hopfield model, Crossformer-SH, ranks first or second, with 30 topand 7 runner-up performances.

**Datasets.** We conduct the experiments on four multivariate time series real-world datasets: ETTh1 (Electricity Transformer Temperature-hourly), ETTm1 (Electricity Transformer Temperature-minutely), WTH (Weather), ILI (Influenza-Like Illness), ECL (Electricity Consuming Load), Traffic.

**Baselines.** We benchmark our method against the results of [Zhang and Yan, 2022] and other baselines (Tranformer [Vaswani et al., 2017], Informer [Zhou et al., 2021] and Autoformer [Chen et al., 2021]) therein.

**Setup.** We adopt the same setting as in [Zhang and Yan, 2022]: multivariate time series prediction task on various datasets. Following [Zhang and Yan, 2022], for each dataset, we evaluate our models with several different prediction horizons. As for hyperparameters, we simply adopt the optimized hyperparameter configuration used in [Zhang and Yan, 2022] obtained via grid search for both [Zhang and Yan, 2022] and all baselines. We report the average accuracy of 5 runs, evaluated using Mean Square Error (MSE) and Mean Absolute Error (MAE) metrics.

### G.5.2 Neural Machine Translation

We showcase the application of the proposed sparse Hopfield model in the context of the classic neural machine translation task, as described in [Vaswani et al., 2017]. By substituting the attention mechanism in the transformer with a 1-step `Hopfield` and `SparseHopfield`, we compare the performance (BLEU score) of the transformer and Hopfield models on various language pairs. The results of this comparison can be found in Appendix G.4.

**Datasets.** We use the WMT17 [Bojar et al., 2017] machine translation task dataset. Which consists of sentence pairs of two different languages, where we consider the translation between German and English (EN-DE), Russian and English (RU-EN). The EN-DE setting has 5.91M pairs of training data, 3000 pairs of validation and 3000 pairs of test data. The EN-RU setting has 25.78M pairs of training data, 3000 pairs of validation and 3000 pairs of test data.

**Baselines.** For the baselines, we follow the architecture of base transformer in [Vaswani et al., 2017] which has 6 layers of encoder and decoder. The hidden dimension is 512 and the feed forward dimension is 2048. More details of configuration can be found in Table 8. Note that when switching the attention in base transformer to either `Hopfield` or `SparseHopfield`, no extra parameter was added in our experiment. Thus, the comparison in our setting is fair.

**Setup.** We follow the setup in [Vaswani et al., 2017] on the WMT-17 dataset. In this experiment, we consider the task of English to German (EN-DE), German to English (DE-EN), Russian to English (RU-EN) and English to Russian (EN-RU). We report the BLEU score on the test set. For WMT17, we follow the base-transformer in [Vaswani et al., 2017], and train the model with 50000 steps, and report the performance of the last checkpoint[12]. Our results show that our proposed `SparseHopfield` not only consistently improves upon transformer-based deep learning models but also surpasses the performance of the dense Hopfield model [Ramsauer et al., 2021].

---

[12]We follow the implementation in https://github.com/OpenNMT/OpenNMT-py for the NMT experiment.

Table 6: **Accuracy comparison for multivariate time series predictions on various datasets, using both the sparse and dense Hopfield models.** Based on SOTA prediction model Crossformer [Zhang and Yan, 2022], we implement two Crossformer variants, **Crossformer-DH** and **Crossformer-SH**, with `Hopfield` and `SparseHopfield` layers respectively. We report the average accuracy of 5 runs, evaluated using Mean Square Error (MSE) and Mean Absolute Error (MAE) metrics. We benchmark our method against the results of [Zhang and Yan, 2022] and other baselines (Transformer [Vaswani et al., 2017], Informer [Zhou et al., 2021] and Autoformer [Chen et al., 2021]) therein. We evaluate each dataset with different prediction horizons (showed in the second column). We have the best results **bolded** and the second best results underlined. In 60+% of 58 settings, the Sparse Hopfield model, Crossformer-SH, ranks first or second, with 30 top and 7 runner-up performances. Our results indicate that our proposed `SparseHopfield` not only consistently enhances transformer-based deep learning models but also achieves SOTA or comparable performance.

| Models | | Transformer | | Informer | | Autoformer | | Crossformer | | Crossformer-DH | | Crossformer-SH | |
|---|---|---|---|---|---|---|---|---|---|---|---|---|---|
| Metric | | MSE | MAE | MSE | MAE | MSE | MAE | MSE | MAE | MSE | MAE | MSE | MAE |
| ETTh1 | 24 | 0.620 | 0.577 | 0.577 | 0.549 | 0.439 | 0.440 | 0.305 | 0.367 | 0.299 | 0.365 | **0.295** | **0.357** |
| | 48 | 0.692 | 0.671 | 0.685 | 0.625 | 0.429 | 0.442 | 0.352 | 0.394 | 0.351 | 0.399 | **0.346** | **0.392** |
| | 168 | 0.947 | 0.797 | 0.931 | 0.752 | 0.493 | 0.479 | **0.410** | **0.441** | 0.412 | 0.443 | 0.425 | 0.455 |
| | 336 | 1.094 | 0.813 | 1.128 | 0.873 | 0.509 | 0.492 | **0.440** | **0.461** | 0.455 | 0.468 | 0.459 | 0.477 |
| | 720 | 1.241 | 0.917 | 1.215 | 0.896 | 0.539 | 0.537 | 0.519 | 0.524 | 0.523 | 0.529 | **0.518** | **0.522** |
| ETTm1 | 24 | 0.306 | 0.371 | 0.323 | 0.369 | 0.410 | 0.428 | 0.310 | 0.371 | 0.199 | 0.285 | **0.198** | **0.287** |
| | 48 | 0.465 | 0.470 | 0.494 | 0.503 | 0.483 | 0.464 | 0.300 | 0.352 | 0.290 | 0.356 | **0.276** | **0.340** |
| | 96 | 0.681 | 0.612 | 0.678 | 0.614 | 0.502 | 0.476 | 0.320 | 0.373 | 0.344 | 0.398 | **0.305** | **0.371** |
| | 288 | 1.162 | 0.879 | 1.056 | 0.786 | 0.604 | 0.522 | 0.404 | 0.427 | 0.404 | 0.429 | **0.373** | **0.406** |
| | 672 | 1.231 | 1.103 | 1.192 | 0.926 | 0.607 | 0.530 | 0.569 | 0.528 | 0.568 | 0.523 | **0.467** | **0.474** |
| WTH | 24 | 0.349 | 0.397 | 0.335 | 0.381 | 0.363 | 0.396 | **0.294** | **0.343** | **0.294** | **0.343** | 0.294 | 0.344 |
| | 48 | 0.386 | 0.433 | 0.395 | 0.459 | 0.456 | 0.462 | 0.370 | 0.411 | **0.369** | **0.408** | 0.375 | 0.410 |
| | 168 | 0.613 | 0.582 | 0.608 | 0.567 | 0.574 | 0.548 | 0.473 | 0.494 | **0.472** | **0.493** | 0.480 | 0.499 |
| | 336 | 0.707 | 0.634 | 0.702 | 0.620 | 0.600 | 0.571 | **0.495** | **0.515** | 0.498 | 0.519 | 0.504 | 0.523 |
| | 720 | 0.834 | 0.741 | 0.831 | 0.731 | 0.587 | 0.570 | **0.526** | **0.542** | 0.528 | 0.546 | 0.536 | 0.544 |
| ILI | 24 | 3.954 | 1.323 | 4.588 | 1.462 | 3.101 | 1.238 | **3.041** | 1.186 | 3.428 | 1.279 | 3.124 | **1.143** |
| | 36 | 4.167 | 1.360 | 4.845 | 1.496 | **3.397** | 1.270 | 3.406 | 1.232 | 3.490 | 1.306 | 3.404 | **1.192** |
| | 48 | 4.746 | 1.463 | 4.865 | 1.516 | **2.947** | **1.203** | 3.459 | 1.221 | 3.600 | 1.277 | 3.509 | 1.205 |
| | 60 | 5.219 | 1.553 | 5.212 | 1.576 | **3.019** | **1.202** | 3.640 | 1.305 | 3.666 | 1.271 | 3.709 | 1.205 |
| ECL | 48 | 0.334 | 0.399 | 0.344 | 0.393 | 0.241 | 0.351 | 0.156 | 0.255 | 0.159 | 0.264 | **0.154** | **0.254** |
| | 168 | 0.353 | 0.420 | 0.368 | 0.424 | 0.299 | 0.387 | 0.231 | 0.309 | 0.290 | 0.316 | **0.225** | **0.303** |
| | 336 | 0.381 | 0.439 | 0.381 | 0.431 | 0.375 | 0.428 | 0.323 | 0.369 | **0.318** | **0.363** | 0.332 | 0.375 |
| | 720 | 0.391 | 0.438 | 0.406 | 0.443 | **0.377** | 0.434 | 0.404 | 0.423 | 0.397 | **0.421** | 0.414 | 0.429 |
| | 960 | 0.492 | 0.550 | 0.460 | 0.548 | **0.366** | **0.426** | 0.433 | 0.438 | 0.434 | 0.438 | 0.440 | 0.443 |
| Traffic | 24 | 0.597 | 0.332 | 0.608 | 0.334 | 0.550 | 0.363 | 0.491 | 0.274 | **0.488** | **0.271** | 0.496 | 0.280 |
| | 48 | 0.658 | 0.369 | 0.644 | 0.359 | 0.595 | 0.376 | 0.519 | 0.295 | **0.513** | 0.291 | 0.516 | **0.290** |
| | 168 | 0.664 | 0.363 | 0.660 | 0.391 | 0.649 | 0.407 | 0.513 | 0.289 | 0.516 | 0.289 | **0.512** | **0.288** |
| | 336 | 0.654 | 0.358 | 0.747 | 0.405 | 0.624 | 0.388 | 0.530 | 0.300 | 0.541 | 0.304 | **0.529** | **0.297** |
| | 720 | 0.685 | 0.370 | 0.792 | 0.430 | 0.674 | 0.417 | 0.573 | 0.313 | 0.557 | 0.307 | **0.555** | **0.304** |

# H  Experimental Details

All experiments are conducted on the platform with NVIDIA GEFORCE RTX 2080 Ti and INTEL XEON SILVER 4214 @ 2.20GHz.

## H.1  Multiple Instance Learning (MIL)

### H.1.1  Synthetic Dataset

**Architectural Details.**   For the Hopfield model, the architecture is composed of 1 layer of either `Hopfield` or `HopfieldPooling` and 1 layer of fully connected output projection. For the attention model, the architecture is composed of 1 layer of attention layer and 1 layer of fully connected output projection. The dataset contains 50% of positive bags and 50% of negative bags.

**Training Details.**   We use an AdamW [Loshchilov and Hutter, 2017] optimizer. For each bag size, we ran the experiment 10 times with different random seed. For all of our synthetic dataset

Table 7: **Results for the machine translation on the WMT17 dataset with language pairs of DE-EN, EN-DE, RU-EN, EN-RU.** We showcase the application of the proposed sparse Hopfield model in the context of the classic neural machine translation task on WMT17 dataset, as outlined in [Vaswani et al., 2017]. By substituting the attention mechanism in the transformer with a 1-step `Hopfield` and `SparseHopfield`, we compare the performance (BLEU score) of the transformer and Hopfield models. To ensure a fair comparison, all models (Transformer, Dense Hopfield, Sparse Hopfield) are of the same size. Our results show that our proposed `SparseHopfield` consistently improves upon transformer-based deep learning models

| Dataset | DE-EN | EN-DE | RU-EN | EN-RU |
|---------|-------|-------|-------|-------|
| Transformer | 29.8 | 34.9 | 28.5 | 24.8 |
| Dense Hopfield | 33.6 | **37.2** | 28.5 | **24.9** |
| Sparse Hopfield | **36.1** | 37.1 | **28.6** | 24.8 |

Table 8: Hyperparameter of the NMT experiment.

| parameter | values |
|-----------|--------|
| batch size | 4096 |
| initial lr | 2.0 |
| vocab size (DE-EN) | 36000 |
| vocab size (EN-RU) | 34776 |
| num heads | 8 |
| hidden dimension | 512 |
| word vector dimension | 512 |
| feed forward dimension | 2048 |
| encoder layer | 6 |
| decoder layer | 6 |
| label smoothing | 0.1 |
| decay method | Noam |
| optimizer | Adam |
| warm up steps (DE-EN) | 4000 |
| warm up steps (EN-RU) | 8000 |
| train steps (DE-EN) | 50000 |
| train steps (EN-RU) | 80000 |
| max sequence length | 96 |
| beam size | 5 |
| tokenizer | sentencepiece |

experiments, we use the exact same configuration, shown in Table 5. The coefficients of Adam optimizer, betas, are set to $(0.9, 0.999)$. As the number of training epochs, we use 150, and the evaluate the model on the testset with the last checkpoint. All the experiments done on the synthetic datasets follow the same architecture and training details.

**Baselines.** For our synthetic dataset, we consider two baselines. [Ilse et al., 2018] (**Gated**), where they proposed a gated-attention mechanism by inserting one extra linear layer on the attention weights before the softmax function. And they replaced the activation function of query to Tanh function. [Vaswani et al., 2017] (**Attn**), where they proposed a multi-head attention mechanism has been widely used in modern deep learning.

Table 9: Statistics of Bit Pattern Synthetic Dataset.

| Unique Patterns | Pattern Length (bits) | Training | Test |
|-----------------|----------------------|----------|------|
| 4 | 4 | 800 | 200 |

Table 10: Hyperparameter of the Bit Pattern Dataset.

| parameter | values |
|---|---|
| batch size | 128 |
| learning rates | $10^{-3}$ |
| scaling factors | 0.25 |
| num heads | 8 |
| head dimension | 8 |
| max update steps | 3 |
| dropout | 0.5 |

### H.1.2 MIL Benchmark Datasets

The experiment is conducted on 4 popular MIL datasets. Elephant, Fox and Tiger are datasets for image annotation which are composed of preprocessed and segmented colored images. Each image is characterized by color, texture and shape descriptors. These datasets contain 100 positive images that contain the purposed animals and 100 negative images that are drawn from a pool of images of other animals. Furthermore, we tested our model on the UCSB breast cancer classification task. An instance in UCSB dataset represents a patch of a histopathological image of cancerous or normal tissue. The detailed statistics of datasets are summarized in Table 11.

Table 11: Statistics of MIL benchmark datasets

| Name | Instances | Features | Bags | +bags | −bags |
|---|---|---|---|---|---|
| Elephant | 1391 | 230 | 200 | 100 | 100 |
| Fox | 1302 | 230 | 200 | 100 | 100 |
| Tiger | 1220 | 230 | 200 | 100 | 100 |
| UCSB Breast Cancer | 2002 | 708 | 58 | 26 | 32 |

In detail, we used a similar architecture described in [Ramsauer et al., 2021] to perform the MIL tasks. Firstly, the instance embeddings are sent to fully connected linear embedding layers with ReLU activation. After that, we used a `SparseHopfield` which has Sparse Retrieval Dynamics to process the output of fully connected linear layers. Afterward, we flatten the output of `SparseHopfield` and use a linear network with ReLU activation can perform classification.

To avoid application bias, we follow the experiment setting in [Küçükaşcı and Baydoğan, 2018, Ramsauer et al., 2021] and utilize a stratified ten-fold cross-validation to demonstrate the success of proposed `SparseHopfield` and `Hopfield`. For each fold in cross-validation, we use a stratified sampling process to split folds for training into a training set and validation set with a 0.1 split rate. We train the models' parameters and tune hyperparameters via grid searching. Once the hyperparameters are selected and the parameters of models are tuned on the first train folds of the first seed, we apply the selected configuration of the model models on other test folds or folds of other random seeds. All reported ROC-AUC scores are the average results of 5 runs with different random seeds.

The grid search space is listed in Table 12. The embedding layers are the pre-HopfieldPooling linear network and the layer width of them is the number of hidden units. A dropout operation, also known as bag dropout, is applied to the attention matrix since it's easy to overfit on these benchmark datasets. All models are trained with the Adam optimizer for 50 epochs. To combat overfitting, we also use an early-stopper that chooses the best checkpoint on the validation set.

One thing that should be noticed is that [Ramsauer et al., 2021] uses the pooling layers `HopfieldPooling` for MIL tasks instead of associative layers `SparseHopfield` or `Hopfield`. We also conduct an ablation experiment in that the model uses the first two modules for MIL tasks following the model structure as well as its training and testing process presented above. As shown in Table 13, the pooling layers can reach comparative results with associative layers on Fox and Tiger datasets but have performance degradation on Elephant and UCSB datasets. Besides, the `SparseHopfieldPooling` can also perform better than `HopfieldPooling` on Tiger and Elephant datasets.

Table 12: Hyperparameter grid search space on the respective validation sets of the Elephant, Fox, Tiger and UCSB breast cancer datasets.

| parameter | values |
|---|---|
| batch size | $\{4, 8, 16\}$ |
| learning rates | $\{10^{-3}, 10^{-5}\}$ |
| learning rate decay | $\{0.98, 0.96, 0.94\}$ |
| embedding layers | $\{1, 2\}$ |
| layer width | $\{32, 64, 128\}$ |
| number of heads | $\{8, 12\}$ |
| head dimensions | $\{16, 32\}$ |
| scaling factors | $\{0.1, 1, 10\}$ |
| bag dropout | $\{0.0, 0.75\}$ |

Table 13: Results for MIL benchmark datsets in terms of AUC score. The models use pooling layers `HopfieldPooling` and `SparseHopfieldPooling` instead.

| Method | Tiger | Fox | Elephant | UCSB |
|---|---|---|---|---|
| w/ `HopfieldPooling` | $0.871 \pm 0.014$ | $0.637 \pm 0.035$ | $0.876 \pm 0.015$ | $0.828 \pm 0.068$ |
| w/ `SparseHopfieldPooling` | $0.884 \pm 0.007$ | $0.610 \pm 0.033$ | $0.914 \pm 0.016$ | $0.796 \pm 0.107$ |