# OpenReview forum: "On Sparse Modern Hopfield Model"
_NeurIPS.cc/2023/Conference — NeurIPS 2023 poster_

### Official Review · Reviewer_usKi · 2023-07-05

**Soundness:** 3 good
**Presentation:** 3 good
**Contribution:** 2 fair
**Rating:** 6
**Confidence:** 3

**Summary:**

The authors introduce sparse Hopfield networks that feature sparse retrieval dynamics corresponding to sparsemax attention mechanisms and resulting in sparse patterns that are more robust to noise. They prove fast convergence analogous to modern Hopfield networks and show how the sparse Hopfield model has a tighter lower bound for memory capacity compared to the dense version. Altered sparse variations of Hopfield layers to be used in Deep Learning models are introduced and their viability is shown on established Image Classification and one synthetic and four real-world Multiple Instance Learning tasks. The novel approach shows increased memory robustness in relation to Gaussian noise applied to the input images.

**Strengths:**

# Significance
The paper's main strength lies in the theoretical results and the Theorems and Lemmas whose proofs can be found in the vast appendix. It's not unlikely that these results will be utilized in future work on associative and biologically plausible Deep Learning. The authors shared their source code facilitating reproducibility.

# Clarity and Quality
The goals of this line of research are presented clearly and summarized into one concise research question. The formatting of math such as definitions and theorems is clear and pleasant to read. Proof sketches are more or less easy to follow. The use of language is of high quality minus a number of typos and grammatically wrong sentences.

**Weaknesses:**

# Originality
The papers originality is fair since sparse computation is a well-known topic in Deep Learning and the presented work is merely an incremental improvement on modern Hopfield networks. The connection between Hopfield networks and attention was made in previous work making the correspondence to a form of sparse attention mechanism an obvious venue of investigation.

# Clarity
The MIL tasks are not introduced very well: The explanation of the bit pattern experiment is insufficient for readers unfamiliar with the task and the Real-world tasks are not explained at all.

# Significance
Potential computation advantages gained from sparse patterns is shortly mentioned in the introduction but not touched upon in the remainder of the manuscript. Additionally, a footnote weakens the claim and puts it into perspective.


**Questions:**

- What is the goal for the MIL tasks?
- Line 192: two-layer fully connected networks -> MLPs?

# Minor mistakes
- Figure and Table 3 are mentioned in the text but are not found in the paper nor in the appendix.
- Line 14: "exploit datasets" is worded strangely
- Same goes for Line 87 "provably blessings"
- Line 112: "memoery"
- Theorem 3.1: Grammatically incorrect
- Line 229: "to utilizes"
- Corollary 3.1.2: "retrieves a memory patterns"
- Line 320: "Boarder impact"

Should the authors decide to fix the given mistakes I am willing to increase my overall rating!

**Limitations:**

The authors do not mention any limitations of their work. However, given the theoretical nature of the work, I do not deem it necessary to discuss them at length. Societal impact of this paper does not surpass the already significant implications of research in the field as a whole.

---

> ### Author Rebuttal · Authors · 2023-08-02
>
> ### The revised draft is in the same Dropbox folder (in Supplementary Material PDF).
> ---
>  >**Reviewer's comment:** The papers originality is fair since sparse computation is a well-known topic in Deep Learning and the presented work is merely an incremental improvement on modern Hopfield networks. The connection between Hopfield networks and attention was made in previous work making the correspondence to a form of sparse attention mechanism an obvious venue of investigation.
>
> **Response:** We acknowledge that the link between Hopfield networks and attention mechanisms has been well-established in prior work. However, In this work, we introduce a principled construction of the energy function through the convex conjugation of entropy regularizer, which sets our work apart from the Modern Hopfield Network (MHN). We consider this our core construction.
>
> Unlike the heuristic approach to the learning rule found in MHN, which is motivated by previous work from 2017, our method provides a more rigorous construction. By using the Gini entropic regularizer, we can analytically derive both the energy function and the retrieval dynamics. The connection to attention is indeed a part of our work, but it is not the main focus. Therefore, in the submitted version, we treated this connection as a remark and included detailed information in the appendix. Our primary emphasis is on the theoretical advancements in the understanding and construction of Hopfield models, which we believe represents a significant step beyond mere incremental improvement.
>
> >**Reviewer's comment:** The MIL tasks are not introduced very well: The explanation of the bit pattern experiment is insufficient for readers unfamiliar with the task and the Real-world tasks are not explained at all.
>
> **Response:**  Multiple Instance Learning (MIL) is a variation of supervised learning where the training set consists of labeled bags, each containing multiple instances.
> The goal of MIL is to predict the bag labels based on the instances they contain, which makes it particularly useful in scenarios where labeling individual instances is difficult or impractical, but bag-level labels are available.
> Examples of such scenarios include medical imaging (where a bag could be an image, instances could be patches of the image, and the label could indicate the presence or absence of disease) and document classification (where a bag could be a document, instances could be the words or sentences in the document, and the label could indicate the topic or sentiment of the document).
>
> Thus, intuitively, attention-based models are suitable for this task. In practice, MIL was widely used in anomaly detection and other tasks that require high cost for fine-level labeling.
>
> >**Reviewer's comment:**  Line 192: two-layer fully connected networks -> MLPs?
>
> **Response:**  We acknowledge that the original wording was unclear.  We have modified it in the latest revision as follows:
> >>SparseHopfieldLayer, by contrast, has learnable memory patterns that maps query patterns to hidden states with sparsemax activation. Thus it can substitute a fully connected layer within deep learning architectures.
>
> Thank you for bringing this to our attention.
>
>
> >**Reviewer's comment:** Figure and Table 3 are mentioned in the text but are not found in the paper nor in the appendix.
>
> **Response:** We appreciate your attention to detail. However, it appears there may have been a slight oversight. Both Figure 3 and Table 3 can be found on page 27 of the appendix in the version of the paper that was submitted.
>
>
> >**Reviewer's comment:** Minor mistakes:  Line 14: "exploit datasets" is worded strangely. Same goes for Line 87 "provably blessings". Line 112: "memoery".  Theorem 3.1: Grammatically incorrect. Line 229: "to utilizes". Corollary 3.1.2: "retrieves a memory patterns" . Line 320: "Boarder impact" .
>
> **Response:**  Thanks for doing such careful proofreading! We have fixed all typos and grammatical errors pointed out.
>
>
> We hope that our responses adequately address the reviewer's concerns, and we look forward to any further feedback. Thank you for your time and consideration.

---

> > ### Comment · Reviewer_usKi · 2023-08-15
> > **Updated overall rating**
> >
> > Thank you for your thorough response to my review!
> >
> > You were able to convince me of the originality of your work. Using the Gini entropy regularizer to derive energy function and retrieval dynamics is undoubtedly a valuable contribution towards understanding the nature of MHNs.
> >
> > Since the authors addressed most of my concerns in the revised version I adjusted my overall rating suggesting acceptance of the manuscript.
> >
> > Best of Luck,
> > Reviewer usKi

---

> > > ### Author Response · Authors · 2023-08-15
> > >
> > > Dear Reviewer usKi,
> > >
> > > We are pleased to hear that our revisions have addressed your concerns, and we are grateful for the time and effort you invested in reviewing our work.
> > >
> > > Your insightful comments have been constructive in enhancing the quality of the draft, especially the careful proofreading.
> > >
> > > Thank you!
> > >
> > > Warm regards,
> > > Authors

---

### Official Review · Reviewer_yZUu · 2023-07-09

**Soundness:** 3 good
**Presentation:** 2 fair
**Contribution:** 2 fair
**Rating:** 4
**Confidence:** 2

**Summary:**

The authors introduce sparse Hopfield model, which are memory-associative models used to store and retrieve patterns. Theoretically, the authors connect sparse Hopfield with sparse attention mechanism and empiricially the authors show how their method can outperform the state-of-the art.

**Strengths:**

The authors present the key research questions, limitations of the current state of the art and their contribution in well-organized sections, detailed theoretic analysis is provided. The figure captions are detailed, and the contributions are clearly stated.

**Weaknesses:**

The paper in terms of language is very hard to follow for layman readers. As Hopfield network may not be familiar to all interested readers, perhaps the authors can give a very basic understanding of what Hopfield network is, why it is interesting and important.
The authors talk about computational efficiency and noise-robustness in the introduction, however experiments comparing efficiency of their method with baseline is missing in current manuscript. How and why sparsity increase noise-robustness is not clearly explained.

**Questions:**

Please see the weakness section.

**Limitations:**

Yes, limitations are discussed.

---

> ### Author Rebuttal · Authors · 2023-08-02
>
> ### The revised draft is in the same Dropbox folder (in Supplementary Material PDF).
> ---
> >**Reviewer's Comment:** The paper in terms of language is very hard to follow for layman readers
>
> **Response:** Thank you for your feedback. We understand the need for making the paper more accessible to readers who may not be familiar with Hopfield networks. To address this, we have included a brief introduction to Hopfield networks in the introductory section of the paper:
>
> >>"Hopfield models are classical associative memory models based on the Ising model, used in both biological and artificial neural networks (Hopfield, 1982; Hopfield, 1984). These models utilize statistical-mechanical retrieval dynamics to store a collection of memory patterns and retrieve the one that is most similar to a given query. For example, if the stored memories are images of all the dogs you've seen in the past, and the query is the image of a dog you see today, the Hopfield model retrieves the memory of the dog that most closely resembles the one you saw today. This is achieved by embedding the memories in the energy landscape of a physical system, where each memory corresponds to a local minimum. When a query is presented, the model initiates energy-minimizing retrieval dynamics at the query, which then navigate the energy landscape to find the nearest local minimum, effectively retrieving the memory most similar to the query."
>
> Moreover, we have made several modifications:
>
> 1. Section 2 now begins with an opening paragraph that simplifies the construction of a well-defined Hopfield model, making it more accessible to non-specialists.
> 2. We have provided an overview of the theoretical results in Section 2.1 to aid readers in developing a deeper understanding of the concepts.
> 3. The discussions on computational benefits have been reformatted into several remarks (Remark 2.3 and 2.4) to improve readability.
> 4. We have added Remark 2.1, which succinctly summarizes the relationship between entropic regularizers and sparse probability mappings, supplemented with relevant references and examples.
> 5. Additional explanations have been included for Definition 2.2 and Lemma 2.2 to enhance clarity.
> 6. Section 4.2 now includes an introductory paragraph about Multiple Instance Learning (MIL).
> 7. For the convenience of the readers, a nomenclature table of notations has been added to the appendix.
>
> We believe these changes will significantly improve the general readability of the draft.
>
> >**Reviewer's Comment:** The authors talk about computational efficiency and noise-robustness in the introduction, however experiments comparing efficiency of their method with baseline are missing in the current manuscript.
>
> **Response:** Thank you for comments. We believe there might have been a slight oversight. The efficiency comparison between our sparse model and the dense baseline can be found in "Convergence Analysis" of Sec. 4.2.2 (summarized as Figure 2). The results indicate that the sparse Hopfield model converges faster than the dense model and also yields superior accuracy.
>
> We acknowledge that our initial submission may not have adequately emphasized the efficiency and noise-robustness. In our revised version, we have made a concerted effort to enhance clarity by breaking down the information into more digestible remarks (remark 2.3 & 2.4).
>
> In the original submission, we mentioned "efficiency" in line 83 of the introduction, referring to the "potential computational efficiency" that our model might inherit from [Martins and Astudillo, 2016]. We also acknowledged in footnote 2 (and Limitations Sec.), that our proposed sparse Hopfield model still faces the issue of quadratic complexity and explained that the term "potential computational efficiency" was meant to highlight the possible efficient implementations of sparsemax that takes advantage of sparsity, as mentioned in Sec. 2 of  [Martins and Astudillo, 2016].
>
> In the revised version, we have clarified our meaning and provided additional context to avoid confusion. Thank you for bringing this to our attention.
>
> On the other hand, there is indeed another kind of "efficiency" advantage provided by our model.
> To clarify, by this "efficiency," we are referring to the efficiency of memory retrieval compared, with dense model. Essentially, a retrieval dynamic with a smaller error converges faster to the fixed points (or stored memories), thereby enhancing efficiency. This concept is elaborated upon in Remark 2.3 of our latest revision.
>
> >>**Remark 2.3 (Faster Convergence)**: Computationally, Theorem 2.2 implies that $\mathcal{T}$ requires fewer updates to reach fixed points with the same amount of error tolerance compared to $\mathcal{T}_{\text{dense}}$. In other words, $\mathcal{T}$ retrieves stored memory patterns faster and therefore more efficiently, as evidenced in Figure 2.
> We have conducted a empirical comparison between the sparse Hopfield model and its dense counterpart in Section 4.1, which is summarized in Figure 2.
>
> >**Reviewer's Comment:** How and why sparsity increase noise-robustness is not clearly explained.
>
> In response to your comment on noise-robustness, Please see line 176-179 of the original submitted version or newly added remark 2.4.
> >>**Remark 2.4 (Noise-Robustness)**: Moreover, in cases of contaminated patterns with noise $\boldsymbol{\eta}$, i.e., $\tilde{\mathbf{x}}=\mathbf{x}+\boldsymbol{\eta}$ (noise in query) or $\tilde{\boldsymbol{\xi}}_\mu=\boldsymbol{\xi}_\mu+\boldsymbol{\eta}$ (noise in memory), the impact of noise $\boldsymbol{\eta}$ on the sparse retrieval error (equation 2.6) is linear, while its effect on the dense retrieval error (equation 2,6) is exponential. This suggests the robustness advantage of the sparse Hopfield model, as evidenced in Figure 1.
>
> Thank you for your time and valuable feedback. Please do not hesitate to let us know if there are any other aspects of our work that you would like us to clarify.

---

### Official Review · Reviewer_XuPM · 2023-07-12

**Soundness:** 2 fair
**Presentation:** 1 poor
**Contribution:** 2 fair
**Rating:** 6
**Confidence:** 2

**Summary:**

This paper proposes the sparse modern Hopfield network. It studies the new proposed model from both perspectives, theoretical and empirical, validating the approach.

**Strengths:**

* The proposed model and the introduction of sparsity in the modern Hopfield network seem novel
* The theoretical analyse is well received
* The empirical validation shows that the sparse model is competitive (if not better) than its dense equivalent
* The code is provided for easy reproducibility

**Weaknesses:**

* The paper is very hard to read. It looks to me that it has a mathematical exposition which is much more complex than necessary, while it fails to develop for the reader basic intuitions about the proposed approach
* The paper structure could have been improved by reserving a fair amount of space to give more details (besides math) about the proposed model and by using any other possible tools (algorithms, illustrations, etc.) to actually show/present how it actually works. Some of the current paper material can be moved into the appendix. This would really ease the reader job.
* I mention that I didn’t follow the math completely, but even so it can be observed that some mathematical notations are not defined, e.g., what is n on line 111?
* The literature survey on related works about sparsity in deep learning is quite weak. I believe that this is relatively important as long as the proposed model aims to be used in deep learning models. Also, the related works discussion shall be in the main paper and not in the appendix.
* The paper needs a careful proofread as it contains typos (for instance, “memoery” on line 112) and English usage can be enhanced for clarity.
* The paper may be potentially impactful in the research community, but given its current state it is arguable (in my opinion) if it can actually become influential

**Questions:**

Q1) Can you please add a nomenclature in the Appendix with all the mathematical notations and symbols? The paper is not an easy read, and a nomenclature would help the reader seriously.

Q2) Can you please discuss and quantify (using various metrics and techniques) the sparsity patterns obtained, besides the relation with sparse attention? How the various sparsity patterns are before, during, or after training? Some (numerical, or visual) examples would help.

**Limitations:**

Fairly well discussed.

---

> ### Author Rebuttal · Authors · 2023-08-04
>
> ### The revised draft is in the same Dropbox folder (in Supplementary Material PDF).
> ---
> >**Reviewer's Comment:** The paper is very hard to read, with a mathematical exposition that seems more complex than necessary. It lacks basic intuitions about the proposed approach.
>
> **Response:** We acknowledge the reviewer's concern and have made revisions to enhance readability. These include:
> - Updating sections, tables, and figures for clarity.
> - Adding a high-level overview of the Hopfield model in the introduction to help readers build intuition.
> - Adding an opening paragraph in Section 2 that explains the construction of a well-defined Hopfield model in layman's terms.
> - Adding an overview of theoretical results in Section 2.1 to help readers build intuition.
> - Adding explanations and visual demonstrations to help readers understand the Sparse Hopfield Model.
> - Including a nomenclature table in the appendix.
> - Rephrasing computational benefits into multiple remarks for readability.
> - Moving broader impacts and future direction into the appendix to save space.
>
> We hope that these modifications have enhanced the overall comprehensibility of the draft.
>
> >**Reviewer's Comment:** The paper structure could be improved by giving more details about the proposed model and using tools like algorithms and illustrations. Some material can be moved to the appendix.
>
> **Response:** We appreciate this suggestion and have made structural changes to include more details and tools to present the model. We have summarized the implementation of the Sparse Hopfield Layer into Algorithm1 (Sec.D.3) in the latest draft. We have also moved some material to the appendix to ease the reader's job.
>
> >**Reviewer's Comment:** Some mathematical notations are not defined, e.g., what is n on line 111?
>
> **Response:** Thank you for pointing this out. We have included a table of notation in the appendix and clarified the definition of n in the latest revision, $n:=||\mathbf{x}||$ is the norm of query $\mathbf{x}$.
>
> >**Reviewer's Comment:** The literature survey on sparsity in deep learning is weak, and the related works discussion should be in the main paper.
>
> **Response:** We appreciate the reviewer's concern. However, we believe a broader survey on "sparsity in DL" is beyond the scope of our work. This work is primarily related to attention and transformers. The current survey, based on the efficient transformer review [Tay 2022]  and many other efficient transformer papers published in the past three years, should suffice. Nonetheless, we are open to specific suggestions on what additional literature should be incorporated.
>
> [Tay 2022] Tay, Yi et al. “Efficient Transformers: A Survey.” ACM Computing Surveys 55 (2020): 1 - 28. (arXiv:2009.06732)
>
> >**Reviewer's Comment:** The paper contains typos and needs proofreading.
>
> **Response:** Thank you for your careful proofreading. We have fixed all identified typos and grammatical errors.
>
> >**Reviewer's Comment:** The paper's potential impact is arguable in its current state.
>
> **Response:** We appreciate the reviewer's concern on the potential impact of our paper.
>
> In response to this, we have taken steps to demonstrate the **practical implications** of our work by implementing the proposed model in **two additional real-world experiments** in the latest revision (Sec. G.5). These experiments utilize sparse Hopfield model for transformer-based models for distinct tasks, including multivariate time series prediction and neural machine translation. The results from these experiments show that our proposed Sparse Hopfield model not only enhances the performance of transformer-based deep learning models consistently, but also achieves state-of-the-art results.
>
> On the **theoretical front**, our work introduces a principled construction of the energy function, achieved through the convex conjugation of the entropy regularizer. This unique approach sets our work apart from the Modern Hopfield Model, marking it as a core contribution of our research. In contrast to the heuristic approach to the learning rule found in MHN, our method provides a more rigorous construction. By utilizing the Gini entropic regularizer, we have been able to analytically derive both the energy function and the retrieval dynamics.
>
> We believe these advancements represent a significant step beyond mere incremental improvement, thereby making a fair contribution to the field.
>
> >**Reviewer's Comment:** Can you please add a nomenclature in the Appendix with all the mathematical notations and symbols?
>
> **Response:** Yes, we have added a nomenclature table in the appendix (Table 3 in the latest revision) to help readers. Thank you for this suggestion!
>
> >**Reviewer's Question:** Q2) Can you please discuss and quantify (using various metrics and techniques) the sparsity patterns obtained, besides the relation with sparse attention? How the various sparsity patterns are before, during, or after training? Some (numerical, or visual) examples would help.
>
>
>
> **Response:** Please see **Figure 2: Quantifying Sparsity in Bit Experiments** in the **attachment of the global rebuttal response** (or the last page of the latest revision).
>
> Here, we quantify and visualize the patterns' sparsity over 5 runs with sparsity metrics (Sparsity Ratio \& Hoyer's Sparsity Measure).
>     We plot the real (target) pattern, initial pattern initialized in dense and sparse $\mathtt{HopfieldPooling}$, and their learned patterns.
>     From above plots, it is easy to see the sparse model obtains sparser patterns than the dense one.
>
> ---
> We sincerely appreciate the time and effort that Reviewer XuPM has invested in reviewing our paper.
> We have taken all comments into careful consideration and have made corresponding revisions to address the concerns raised.
>
> We hope that the revisions and clarifications provided in this response address the reviewer's concerns and make the value of our work clear. We look forward to further feedback and discussion.

---

> > ### Comment · Reviewer_XuPM · 2023-08-18
> > **Rating updated**
> >
> > I thank the authors for carefully considering all my comments and for the extensive rebuttal in general. Really appreciating it. After reading the other reviews and the authors' answers, I have increased my rating to weak accept.

---

> > > ### Author Response · Authors · 2023-08-18
> > > **Thank You for Constructive Comments**
> > >
> > > Dear Reviewer XuPM,
> > >
> > > We're happy that our revisions have met your expectations. We truly appreciate your thoughtful feedback throughout the review process! Your insights have been invaluable in refining our paper. Thank you!
> > >
> > > Best regards,
> > >
> > > Authors

---

### Official Review · Reviewer_haSG · 2023-07-27

**Soundness:** 3 good
**Presentation:** 3 good
**Contribution:** 3 good
**Rating:** 6
**Confidence:** 3

**Summary:**

The paper proposes a new model from the Dense Associative Memory family that uses Sparsemax function for its energy. This model is studied and compared to the model with the softmax-based energy from both theoretical and empirical perspectives. The work proposes that the sparse model outperforms the model with the softmax (dense model) in terms of the memory retrieval bound.

**Strengths:**

The work derives a novel model from the modern HN family and analyses its energy and dynamics using convex conjugate of the sparse entropic regularizer. As far as I can tell this is a sophisticated result, which meaningfully extends the family of models that has been previously studied.

The work studies theoretically several properties of the convergence dynamics in this new model. In general, results pertaining to continues models from this family are scarce, which makes this submission even more valuable.

The authors also discuss possible ways of integrating their model into existing Hopfield-like frameworks, and propose Sparse Hopfield, SparseHopfieldPooling, SparseHopfieldLayer layers based on their energy function.

Empirical results look promising.

The proofs in the appendices look convincing, although I have not checked them carefully.

**Weaknesses:**

It would be nice if the authors could summarize in a concise and crisp way what is the theoretical advantage of their model with the Sparsemax compared to previously studied modern HN models. I can see that the new model uses a very different language for its formulation (which is great), but I am struggling to understand its computational benefits compared to previously studied models. The improvements in empirical performance, which the authors present are great, but they are not too significant to claim superiority just based on them. I would appreciate a clear theoretical proposition here.

The statement in lines 578-579 is somewhat confusing. The model studied in Ramsauer 2020 uses softmax activation function, the model studied in Demircigil uses exponential activation function. Several papers (incorrectly) state that these two models are identical. This is wrong, since they have mathematically distinct energy functions and update equations.

Some wording in Appendix C might be somewhat confusing. The appendix makes it sound that everyone before Ramsauer 2020 studied binary Hopfield networks, but Ramsauer 2020 introduced the continuous networks. This is not quite correct. For instance https://www.pnas.org/doi/10.1073/pnas.81.10.3088 introduced continues sparse (as opposed to dense) Hopfield networks in 1984. Krotov & Hopfield 2016 introduced continuous dense Hopfield networks (see equation 10 in their paper and most of the empirical results on MNIST), etc. Ramsauer 2020 focused on studying a specific model from that family (with softmax activation) and calculated the capacity of that specific model. But the continuous networks (both sparse and dense) were introduced in prior work.


**Questions:**

1. Could the authors please explain the computational benefits of their model compared to previously studied dense associative memories?

2. I am not too familiar with Gini entropy and its conjugate, but formulas for the relationship between $F(\mathbf{p})$ and $\Psi(\mathbf{p})$ (e.g. above equation 2.4) look very similar to Legendre transformation to me. Krotov & Hopfield 2020 also use Legendge transformation to compute the energy function using Lagrangians. Is there a precise relationship between the Lagrangian language of Krotov and Hopfield and the formalism developed in this submission?

3. The authors present quantitative metrics to empirically evaluate the performance of their model. This is great, but in order to get more intuition about the new model it would be helpful to show a few images for experiments presented in section 4.1 that the authors used in their experiments. For example pairs of initial states and final retrieved images. It would be interesting to see visually what kinds of mistakes the new network makes compared to previously studied models.


**Limitations:**

This is a theoretical work. The authors address societal impacts in the last section “Broader Impact” and appendix A.

---

> ### Author Rebuttal · Authors · 2023-08-02
>
> ### The revised draft is in the same Dropbox folder (in Supplementary Material PDF).
> ---
> > Q1 & It would be nice if the authors could summarize in a concise and crisp way...
>
> **Response**: Thank you for your comment. Please see below summary of the advantages of our sparse model over its dense counterpart.
>
> 1. **Principled Derivation of Energy Function and Retrieval Dynamics (Sec. 2.1 & 2.2)**: Unlike the heuristic design of the energy function in the dense modern Hopfield model, we provide a principled derivation for the energy function and the corresponding retrieval dynamics. This is based on the insight that the convex conjugate of various entropic regularizers can result in distributions exhibiting different levels of sparsity.
>
> 2. **Faster Convergence of Memory Retrieval (Remark 2.3 in the latest version)**: Our model demonstrates faster convergence of memory retrieval from the tighter retrieval error bound. The sparser the pattern is, the faster the convergence.
> >**Remark 2.3 (Faster Convergence)**: Computationally, Theorem 2.2 implies that $\mathcal{T}$ requires fewer updates to reach fixed points with the same amount of error tolerance compared to $\mathcal{T}_{\text{dense}}$. In other words, $\mathcal{T}$ retrieves stored memory patterns faster and therefore more efficiently, as evidenced in Figure 2.
>
>
> 3. **Noise-Robustness (Remark 2.4 in the latest version)**: Our model exhibits significant advantages in scenarios with higher noise levels, as shown in Fig. 1 of the experimental section.
> >**Remark 2.4 (Noise-Robustness)**: Moreover, in cases of contaminated patterns with noise $\boldsymbol{\eta}$, i.e., $\tilde{\mathbf{x}}=\mathbf{x}+\boldsymbol{\eta}$ (noise in query) or $\tilde{\boldsymbol{\xi}}_\mu=\boldsymbol{\xi}_\mu+\boldsymbol{\eta}$ (noise in memory), the impact of noise $\boldsymbol{\eta}$ on the sparse retrieval error (equation 2.6) is linear, while its effect on the dense retrieval error (equation 2,6) is exponential. This suggests the robustness advantage of the sparse Hopfield model, as evidenced in Figure 1.
>
>
> 4. **Tighter, Sparsity-Dependent Well-Separation Condition (Thm. 3.1)**: Intuitively, the sparser the pattern is, the easier it is for the well-separation condition to be fulfilled. This means it is easier to separate and retrieve stored memories. The hardness of distinguishing patterns can be tamed by the sparsity, preventing an increase of separation $\Delta_\mu$ with $M$ as observed in the dense Hopfield model.
>
> The benefits of sparsity arise from the increased energy landscape separation provided by the sparse Hopfield energy function. This enables the separation of closely correlated patterns, resulting in a tighter well-separation condition for distinguishing such patterns.
>
> We hope these points clarify the theoretical advantages of our model.
>
>
> >**Reviewer's Comment:** The improvements in empirical performance, which the authors present are great, but they are not too significant to claim superiority just based on them
>
> **Response:** In response to this, we have added two additional real-world experiments showing that our proposed Sparse Hopfield model not only enhances the performance of transformer-based deep learning models consistently, but also achieves STOA results.
>
>
> >**Reviewer's Comment:**
> The statement in lines 578-579 is somewhat confusing....
>
> **Response:**
> We appreciate your attention to this detail, and we fully agree with your observation. The models referenced are indeed distinct, and  many existing works state their resemblance incorrectly.
>
> Yet, our statement that “[Ramsauer 2020] generalizes the energy function of [Demircigil 2017]” was intended to highlight the relationship between the **energy functions**, not the activation functions. We did recognize that [Demircigil 2017] does not explicitly define activation functions, and its retrieval dynamics are not related to attention at all.
>
> We acknowledge that the wording in lines 578-579 may have led to confusion. To ensure clarity and accuracy, we have revised the paragraph in question. Thank you for bringing this to our attention.
>
> >**Reviewer's Comment:**
> Some wording in Appendix C might be somewhat confusing...
>
> **Response:**
> We appreciate your clarification. Our intention was not to portray the Modern Hopfield model as the first continuous Hopfield model. We understand how our wording in Appendix C could have led to confusion, and we have revised it accordingly. We have also expanded our citations to include more of the relevant prior work. Please refer to footnote 9 in the updated version of our paper for these changes. Thank you!
>
> >Q2 & I am not too familiar with Gini entropy and its conjugate...
>
> **Response:**
> Your observation is astute. Indeed, the relationship between Gini entropy and its conjugate, as well as the formulas we've presented (such as equation 2.4), do bear a resemblance to the Legendre transformation.
>
> Specifically, if we let `h_μ = x, Ξ` as per eqn 5 of [Krotov & Hopfield 2020], and set `L_h = Ψ*` such that `∇Ψ* = sparsemax = f_μ` and `L_v = 1/2 x^2` such that `∇L_v = v_i`, we can rewrite our proposed energy function in their formulation, up to an additive constant. Furthermore, the finite difference version of their update rule aligns with our retrieval dynamics.
>
> It's clear that `L_h` corresponds to the convex conjugate of entropic regularizers. While the formulation in Krotov & Hopfield 2020 necessitates a degree of hindsight to derive different Hopfield variants, our methodology provides a systematic and principled approach for deriving `L_h` for models exhibiting varying degrees of sparsity.
>
> >Q3: The authors present quantitative metrics to empirically evaluate the performance of their mode...
>
> Please see the Sec. H.1 (Visualization of Experimental Validation of Theoretical Results) of revised draft.
>
> We're open to any further questions or clarifications you might have about our work.

---

> > ### Comment · Reviewer_haSG · 2023-08-17
> > **Thank you for answering my questions and clarifications.**
> >
> > Dear Authors,
> >
> > thanks for answering my questions and all the clarifications. I think I understand the paper better now.

---

> > > ### Author Response · Authors · 2023-08-17
> > > **Thank You for Insightful Review**
> > >
> > > Dear Reviewer haSG,
> > >
> > > We are happy to hear that our revisions have addressed your concerns.
> > >
> > > Thank you again for your constructive comments, which are pivotal in improving our draft and presenting a clearer view of the modern Hopfield model family. We truly appreciate your thorough review.
> > >
> > > Best,
> > >
> > > Authors

---

### Author Rebuttal · Authors · 2023-08-08

Dear Reviewers,

We thank the reviewers for the insightful questions and reviews. We have answered all the questions and addressed all the problems in detail in rebuttal and revision.

The latest revision is readily available in the reproducibility code folder. Any changes or modifications made from the submitted version are highlighted in blue in this updated draft.

In response to the reviewers' suggestions, we have made revisions to improve the overall readability of the paper.
We have conducted another round of proofreading and have fixed all typos.
Several sections, tables, and figures have been updated for clarity and completeness. These revisions include additional explanations, paragraphs and sections to help readers build intuition, and visual demonstrations of the retrieval task to highlight the advantages of the sparse model over its dense counterpart. Most importantly, a new section, Section G.2 in the appendix, has been added to present **two practical scenarios of integrating the Sparse Hopfield Model into transformer-based deep learning models**.

---
## Revision Details (The revised draft is in the same Dropbox folder in Supplementary Material PDF)

**Major revisions include:**

-  **Two Real-World Additional Experiments**:
   - We have implemented the proposed model in two additional experiments that utilize transformer-based models for distinct tasks.
   - These tasks include multivariate time series prediction and neural machine translation.
   - Our results demonstrate that our proposed Sparse Hopfield model consistently enhances transformer-based deep learning models and achieves state-of-the-art performance.
    - In **60+% of 58 settings, the Sparse Hopfield model ranks first or second**, with 28 top and 7 runner-up performances in the time series prediction task.
    - [Table 1 and Table 2 of the **attachment** or Sec. G.2 of the revised draft]
- **Visual demonstration of retrieval:** We provide a visual demonstration of the advantages of the sparse Hopfield model over its dense counterpart.
    - [Figure 1 (**Visualizing Noise-Robustness of Sparsemax and Softmax Hopfield Models**) of **attachment** or Figure 4 in  Sec.G.1 of the revised draft]
- Additional evaluation and visualization of sparse patterns.
    - [Figure 2 (**Quantifying Sparsity in Bit Experiments**) of **attachment**]
- An opening paragraph in Section 2 that explains the construction of a well-defined Hopfield model in a manner that is easily understandable to non-specialists.
- An overview of theoretical results in sec2.1 to help readers build intuition.
-  A **nomenclature table** of notations in the appendix.
    - [Table 3 in Sec.A of the revised draft]
- An algorithm summarizing the implementation of the sparse Hopfield layer with multiple updates.
    - [Algorithm 1 in Sec.D.3 of the revised draft]

**Minor revisions include:**

- Proofreading the manuscript and fixing all identified typos and grammatical errors by reviewers and authors.
- Moving the broader impacts and future direction paragraph into the appendix to save space.
- Rephrasing paragraphs regarding computational benefits into multiple remarks for enhanced readability. [remarks 2.3, 2.4 (line 224-231) of the revised draft]
- Adding a remark summarizing the relationship between entropic regularizers and sparse probability mappings, with citations and examples. [remark 2.1 (line 154-157) of the revised draft]
- Rephrasing the remark below Lemma 2.1 to include intuition about $\beta$. [Remark 2.2  (line 188-193) of the revised draft]

We hope these revisions address the reviewers' concerns and improve the overall quality of our paper.

---
**What are included in the attachment:**
1. Figure 1: **Visualizing Noise-Robustness of Sparsemax and Softmax Hopfield Models** (Section 4.1 & Figure 4).
2. Table 1: **Additional Real-World Experiment: Multivariate Time Series Prediction** (Appendix G.5.1).
3. Figure 2: **Quantifying Sparsity in Bit Experiments** (Section 4.2.1).
4. Table 2: **Additional Real-World Experiment: Machine Translation on the WMT17 Dataset with Language Pairs of DE-EN, EN-DE, RU-EN,
EN-RU** (Appendix G.5.2)

---
**The Nomenclature Table:** [Sec. A of the revised draft]

| Symbol | Description |
|---|---|
| $\langle\mathbf{a},\mathbf{b}\rangle$ | Inner product for vectors $\mathbf{a},\mathbf{b}\in \mathbb{R}^d$ |
| $[I]$ | Index set $\{1,\cdots,I\}$, where $I\in\mathbb{N}^+$ |
| $\|\cdot\|_2$ | Spectral norm, equivalent to the $l_2$-norm when applied to a vector |
| $d$ | Dimension of patterns |
| $M$ | Number of stored memory patterns |
| $\beta$ | A scaling factor of the energy function that controls the learning dynamics |
| $\mathbf{x}$ | State/configuration/query pattern in $\mathbb{R}^d$ |
| $\mathbf{\xi}$ | Memory patterns (keys) in $\mathbb{R}^d$ |
| $\mathbf{\Xi}$ | Shorthand for $M$ stored memory (key) patterns in $\mathbb{R}^{d\times M}$ |
| $\mathbf{\Xi}^T \mathbf{x}$ | $M$-dimensional overlap vector |
| $[\mathbf{\Xi}^T \mathbf{x}]_\kappa$ | The $\kappa$-th element of  $\mathbf{\Xi}^T \mathbf{x}$ |
| $n$ | Norm of $\mathbf{x}$, denoted as $n\coloneqq\|\mathbf{x}\|$ |
| $m$ | Largest norm of memory patterns, denoted as $m\coloneqq \max_{\mu\in[M]}\|\mathbf{\xi}_\mu\|$ |
| $\kappa$ | The number of non-zero element of Sparsemax |
| $R$ | The minimal Euclidean distance across all possible pairs of memory patterns |
| $S_\mu$ | The sphere centered at the memory pattern $\mathbf{\xi}_\mu$ with finite radius $R$ |
| $\mathbf{x}^\star_\mu$ | The fixed point of $\mathcal{T}$ covered by $S_\mu$ |
| $\Delta_\mu$ | The separation of a memory pattern $\mathbf{\xi}_\mu$ from all other memory patterns $\mathbf{\Xi}$ |
| $\tilde{\Delta}_\mu$| The separation of $\mathbf{\xi}_\mu$ at a given $\mathbf{x}$ from all memory patterns $\mathbf{\Xi}$ |

---

### Decision · Program_Chairs · 2023-09-21

**Decision:**

Accept (poster)

**Comment:**

The paper makes some very nice contributions regarding Hopfield networks, by proposing a sparse version preserving the desirable properties of its dense counterpart w.r.t. convergence and capacity, as well as connection with attention mechanisms.

The authors provided substantial feedback to the reviews, which satisfactorily addressed key concerns.

We strongly urge the authors to incorporate their feedback materials into the manuscript. In particular the additional experiments and visualizations add greatly to the significance of the submission.